# Age-associated changes to neuronal dynamics involve a disruption of excitatory/inhibitory balance in *C. elegans*

Gregory S Wirak[1], Jeremy Florman[2], Mark J Alkema[3], Christopher W Connor[4,5], Christopher V Gabel[6]*

[1]Graduate Program for Neuroscience, Department of Physiology and Biophysics, Boston University, Boston, United States; [2]Department of Neurobiology, University of Massachusetts Medical School, Worcester, United States; [3]University of Massachusetts Medical School, Worcester, United States; [4]Department of Anesthesiology, Perioperative and Pain Medicine, Brigham and Women's Hospital, Boston, United States; [5]Research Associate Professor of Physiology and Biophysics, Boston University School of Medicine, Boston, United States; [6]Department of Physiology and Biophysics, Neurophotonics Center, Boston University School of Medicine, Boston, United States

**Abstract** In the aging brain, many of the alterations underlying cognitive and behavioral decline remain opaque. *Caenorhabditis elegans* offers a powerful model for aging research, with a simple, well-studied nervous system to further our understanding of the cellular modifications and functional alterations accompanying senescence. We perform multi-neuronal functional imaging across the aged *C. elegans* nervous system, measuring an age-associated breakdown in system-wide functional organization. At single-cell resolution, we detect shifts in activity dynamics toward higher frequencies. In addition, we measure a specific loss of inhibitory signaling that occurs early in the aging process and alters the systems' critical excitatory/inhibitory balance. These effects are recapitulated with mutation of the calcium channel subunit UNC-2/CaV2α. We find that manipulation of inhibitory GABA signaling can partially ameliorate or accelerate the effects of aging. The effects of aging are also partially mitigated by disruption of the insulin signaling pathway, known to increase longevity, or by a reduction of caspase activation. Data from mammals are consistent with our findings, suggesting a conserved shift in the balance of excitatory/inhibitory signaling with age that leads to breakdown in global neuronal dynamics and functional decline.

## Editor's evaluation

Wirak and colleagues record single cell resolution whole brain dynamics in ageing *C. elegans*. They make the intriguing observations that coordination of brain wide neuronal activity dynamics declines with age, associated with reduced negative correlativity indicating a shift in the excitatory-inhibitory balance across the brain.

*For correspondence:
cvgabel@bu.edu

Competing interest: The authors declare that no competing interests exist.

## Introduction

Progressive breakdown of neuronal function is a hallmark of natural aging. Yet the relationship between the molecular, cellular, and ultrastructural modifications (*Mattson and Arumugam, 2018*), the alterations in neuronal dynamics and signaling, and the deficits in cognitive performance and behavior that emerge with advanced age remain unclear. At the cellular level, dramatic neuronal loss is not observed; however, changes in neuronal excitability and signaling have been well documented (*Peters et al., 2008*; *Morrison and Baxter, 2012*). These effects include decreased inhibition within multiple sensory systems (*Richardson et al., 2013*; *Schmidt et al., 2010*; *David-Jürgens and Dinse, 2010*; *Cheng and Lin, 2013*) and the hippocampus (*Potier et al., 2006*), suggesting a potential shift in the balance of excitatory/inhibitory signaling with age. Maintenance of this signaling balance is critical for functional homeostasis in the nervous system and its breakdown is associated with numerous pathological states, including autism spectrum disorders, epilepsy, and Alzheimer's disease (*Turrigiano and Nelson, 2004*; *Isaacson and Scanziani, 2011*; *Vico Varela et al., 2019*). While such imbalances might also contribute to shifts in functional organization with age, the link between alterations in cellular signaling and breakdown of system dynamics is ill-defined. Recent studies employing task-free functional magnetic resonance imaging have shown that aging alters intrinsic brain connectivity and the functional organization of large-scale resting state networks (*Sala-Llonch et al., 2015*), which correlate with diminished cognitive performance (*Bagarinao et al., 2019*; *Varangis et al., 2019*). To further elucidate the link between age-associated decline in system-state dynamics and changes in activity and signaling at the cellular level, we turned to comprehensive system-wide functional neuronal imaging within the nematode *Caenorhabditis elegans*.

Recent advances in multi-neuron fluorescence imaging present an unprecedented ability to measure and understand the dynamics, function, and breakdown of small neuronal systems. Applied to the microscopic nematode *C. elegans,* such techniques enable measurement of neuronal activity across the majority of the nervous system with single-cell resolution, revealing spontaneous behavioral state dynamics (*Kato et al., 2015*; *Nguyen et al., 2016*; *Venkatachalam et al., 2016*) that are lost under unique circumstances, such as developmental quiescence or exposure to volatile anesthetics (*Nichols et al., 2017*; *Awal et al., 2020*). Moreover, *C. elegans* is also a powerful system for the study of aging with longevity studies contributing much to our understanding of conserved molecular pathways affecting organismal aging (*Mack et al., 2018*). *C. elegans* is both short-lived and genetically tractable, facilitating the high-throughput analyses of how these pathways and their manipulation affect various aspects of normal aging. With its capabilities for comprehensive multi-neuron imaging and well-established aging studies, *C. elegans* presents a unique window into the age-associated breakdown of neuronal signaling and system dynamics.

Neuronal aging studies in *C. elegans* illustrate that its nervous system, comprised 302 stereotyped and identifiable neurons, is largely spared from large anatomical age-related deterioration, paralleling what is seen in higher organisms. Gross morphology is well preserved, with limited neurite deterioration (*Herndon et al., 2002*), no neural apoptosis or necrosis (*Pan et al., 2011*), and no significant changes to the nuclear architecture (*Haithcock et al., 2005*). Closer inspection of neurite ultrastructure, however, revealed novel age-dependent features, including ectopic outgrowths and the beading and blebbing of axons (*Pan et al., 2011*; *Tank et al., 2011*; *Toth et al., 2012*). A breakdown in synaptic integrity was also found to accompany aging, with evidence suggesting that functional neuronal decline precedes that of the musculature. Initial shifts in behavior, several of which correlate with reduced synaptic integrity, may therefore reflect changes in the nervous system rather than sarcopenia, although disentangling the two has been challenging (*Toth et al., 2012*; *Liu et al., 2013*; *Mulcahy et al., 2013*). The disruption of specific behaviors in the context of normal worm aging has also been well documented and includes behaviors ranging from defecation (*Croll et al., 1977*; *Bolanowski et al., 1981*; *Felkai et al., 1999*), crawling dynamics (*Glenn et al., 2004*; *Podshivalova et al., 2017*), chemotaxis (*Glenn et al., 2004*), to associative learning (*Kauffman et al., 2010*). To date, however, studies that have directly examined neuronal activity underlying such behavioral changes have been limited to individual neurons and synapses (*Chokshi et al., 2010*; *Mulcahy et al., 2013*; *Liu et al., 2013*; *Zullo et al., 2019*; *Huang et al., 2020*) or simple circuits (*Leinwand et al., 2015*).

To bridge the gap between age-associated alterations at the cellular level and concomitant deterioration in neuronal function and behavior, we perform comprehensive functional multi-neuron fluorescence imaging in *C. elegans* throughout its lifespan. As the animals age, we measure a distinct loss of

inhibitory signaling that alters the excitatory/inhibitory balance of the nervous system and is accompanied by an increase in individual neuron activity. These cellular effects correspond with a breakdown of system-wide behavior state dynamics. We determine that these effects are recapitulated by modifications of the calcium channel subunit UNC-2/CaV2α known to alter excitatory/inhibitory balance in *C. elegans* (*Huang et al., 2019*) or by changes in inhibitory GABA signaling. In addition, we show that a long-lived mutant background delays many aspects of normal neuronal aging as does loss of caspase activation. Our findings in *C. elegans* parallel numerous aspects of neuronal aging found in higher organisms, emphasizing the loss of inhibitory signaling and disruption of excitatory/inhibitory balance as a key element of neuronal decline and begin to uncover the cellular mechanisms driving these changes.

## Results

### Command interneuron AVA becomes more active but exhibits slower transition dynamics with age

As the nematode *C. elegans* ages, the locomotive behavior of the animal is grossly altered (*Figure 1a*). Young adult worms tend to crawl in a forward direction with infrequent bouts of spontaneous backward movement (reversals), which typically precede changes in the direction of travel and constitute an important component of the animal's foraging strategy (*Pierce-Shimomura et al., 1999*; *Croll, 1975a*; *Croll, 1975b*). As has been previously shown (*Glenn et al., 2004*; *Podshivalova et al., 2017*), we measure a progressive increase in the rate at which animals initiate reversals with age (*Figure 1b*). We further observe that the typical reversal becomes shorter, resulting in the aged animal making more brief or 'hesitant' reversals. While a reversal can be elicited by anterior mechanosensation (*Chalfie et al., 1985*), the age-dependent increase in reversal frequency cannot be explained by an enhancement in anterior touch sensitivity. Instead, we measure a decrease in touch sensitivity with age (*Figure 1c*), consistent with previous reports (*Vayndorf et al., 2016*). Therefore, this behavioral shift likely arises due to age-related changes in neuronal dynamics within the neurocircuitry controlling reversal behavior.

To determine the effects of aging on the command interneurons controlling *C. elegans* crawling, we performed functional fluorescence imaging of the premotor interneuron AVA, known to mediate backward locomotion (*Chalfie et al., 1985*; *Kawano et al., 2011*; *Figure 1d*). AVA calcium transient traces measured as GCaMP fluorescence (*Figure 1e*) were acquired from immobilized worms spanning the animal's adult life, that is days 1, 3, 6, 9, and 12 of adulthood. In healthy young animals, the AVA interneuron displays binary dynamics as it switches between active and inactive states. These states have been shown to correlate closely with reverse and forward crawling behavior, respectively (*Chalfie et al., 1985*; *Kawano et al., 2011*). In aged worms, a striking reduction was observed in the rate at which AVA transitions from inactive to active state (i.e. from low to high GCaMP6s fluorescence). This is made visually apparent by overlaying and averaging the individual calcium transient onsets (*Figure 1f*). Transient rise times were calculated as the time required to shift from 5 to 95% of the maximum fluorescence (as previously described [*Wirak et al., 2020*], see Statistical Methods). A progressive increase in AVA rise time is observed through days 1–12 of nematode adulthood (*Figure 1g*). In contrast, the fall time (the time required to shift from 95 to 5% of the maximum fluorescence) shows no significant change with age (*Figure 1f and h*). Notably, the proportion of time AVA exhibits high activity (as quantified by the duty ratio of the signal) progressively increases with age (*Figure 1i*). This is primarily due to an increase in calcium transient duration, while the frequency of calcium transients remains unchanged (*Figure 1—figure supplement 1*). Increased AVA activity is consistent with the increased tendency for reversals in aged animals. However, it should be noted that the neural activity patterns of immobilized *C. elegans* differ from those of freely behaving animals (*Kato et al., 2015*), making it difficult to draw direct comparisons between the two contexts. Specifically, freely moving animals display short-lived and spikey AVA calcium transients, as opposed to the sustained periods of high activity observed in immobilized animals.

### Global neuronal activity becomes increasingly disorganized with age

We next sought to assess system-wide neuronal dynamics across the *C. elegans* lifespan. Using methods we recently developed, we employed light sheet microscopy to simultaneously capture the

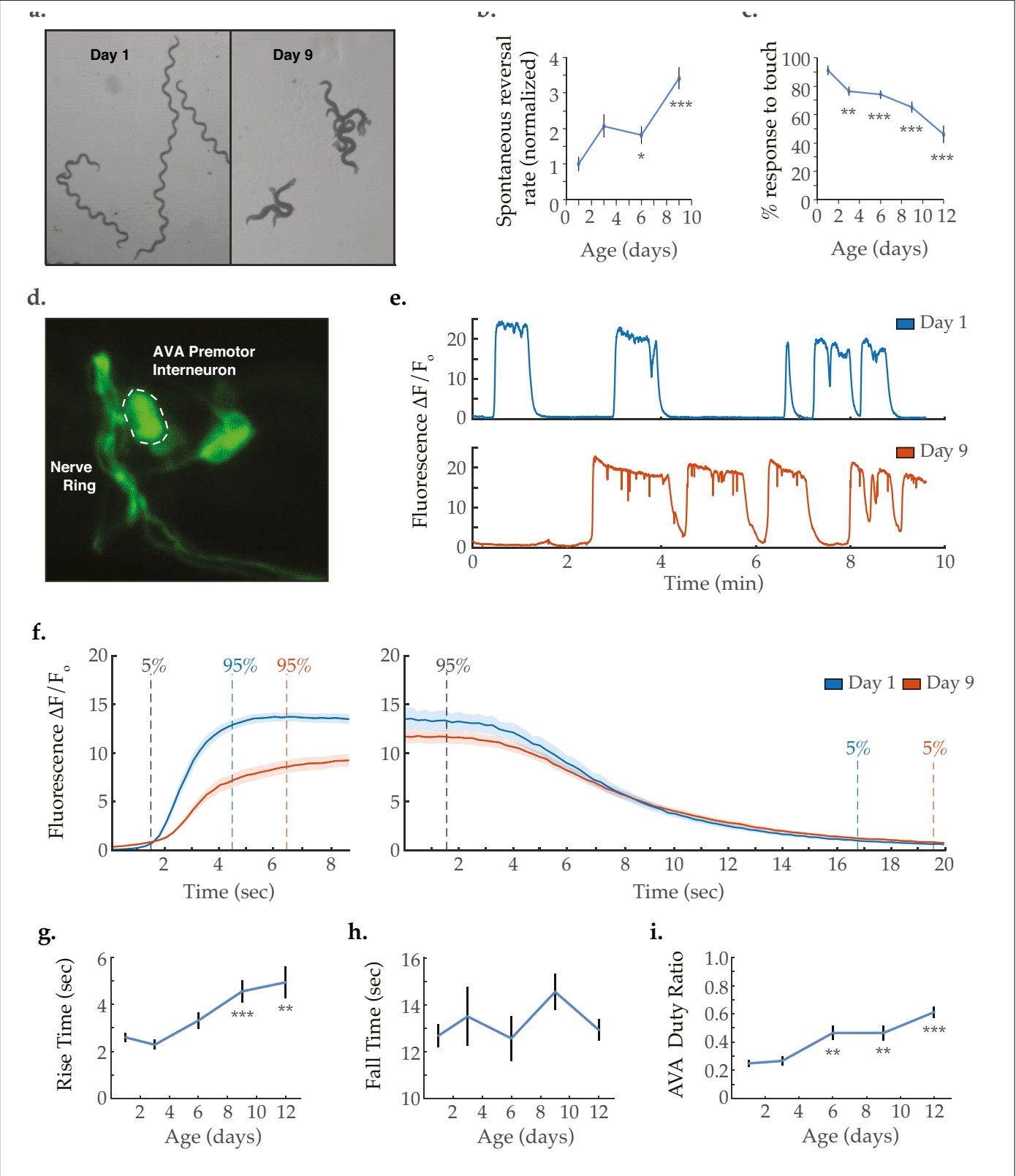

**Figure 1.** Activity dynamics of the AVA command interneuron with age. (**a**) Travel paths of two and three *Caenorhabditis elegans*, respectively, on day 1 and day 9 of adulthood (over 30 s). (**b**) Spontaneous reversal rates of adult QW1574 strain worms of various ages, normalized to day 1 frequency. (For days 1, 3, 6, and 9, n=9, 18, 22, and 14 animals, respectively). (**c**) Anterior touch responsiveness in adult QW1574 strain worms of various ages. Responsiveness denotes how often an eyelash drawn across the anterior portion of an animal's body elicited a reversal (see Materials and methods).

*Figure 1 continued on next page*

*Figure 1 continued*

(For days 1, 3, 6, 9, and 12, n=20, 44, 53, 53, and 33 animals, respectively). (**d**) Transgenic expression of the calcium-sensitive green fluorescent protein GCaMP6s in the command interneuron AVA. (**e**) GCaMP calcium transients measured from the AVA interneuron in immobilized animals at days 1 and 9 of adulthood. (**f**) Average AVA GCaMP fluorescence transient onsets and offsets in day 1 and day 9 worms (shaded areas delineate the standard error of the mean). For day 1 and day 9 onsets, n=37 and 38 traces from 20 and 16 animals, respectively. For day 1 and day 9 offsets, n=33 and 32 from 19 and 14 animals, respectively. (**g, h**) The average rise time (i.e. transient time from an 'ON' transition from 5 to 95% of maximum fluorescence) and average fall time (95–5%) of AVA GCaMP fluorescence transients at various ages. For days 1, 3, 6, 9, and 12 onsets, n=37, 34, 36, 38, and 43 traces from 20, 13, 13, 16, and 14 animals, respectively. For days 1, 3, 6, 9, and 12 offsets, n=33, 24, 28, 32, and 29 traces from 20, 11,11, 14, and 13 animals, respectively. (**i**) Duty ratio of AVA fluorescence, defined as the proportion of time that AVA is active versus inactive (see Materials and methods). For days 1, 3, 6, 9, and 12, n=21, 13, 13, 12, and 16 animals, respectively. Error bars denote standard error of the mean. *p<0.05; ** p<0.01; *** p<0.001, Sidak post hoc test.

The online version of this article includes the following source data and figure supplement(s) for figure 1:

**Source data 1.** *C. elegans* reversal behavior and anterior mechanosensation with age.

**Source data 2.** AVA interneuron activity state transition dynamics with age.

**Source data 3.** AVA interneuron calcium transient frequency, duration, and duty ratio with age.

**Figure supplement 1.** AVA calcium transient frequency and duration.

GCaMP6s signal from 120 neurons within the *C. elegans* head region (*Awal et al., 2020*). Measurements were acquired from worms at days 1, 3, 6, 9, and 12 of adulthood with 10–20 animals per condition. Experiments focused on day 9 as a robust senescent time point. Example videos of head region neural activity in worms at days 1 and 9 are displayed as two-dimensional (2D) projections in *Videos 1 and 2*, with the location of each neuron identified by our tracking software (see Materials

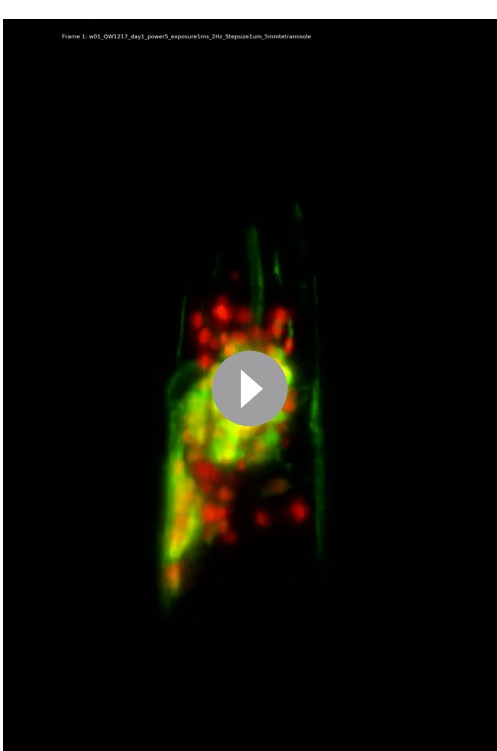

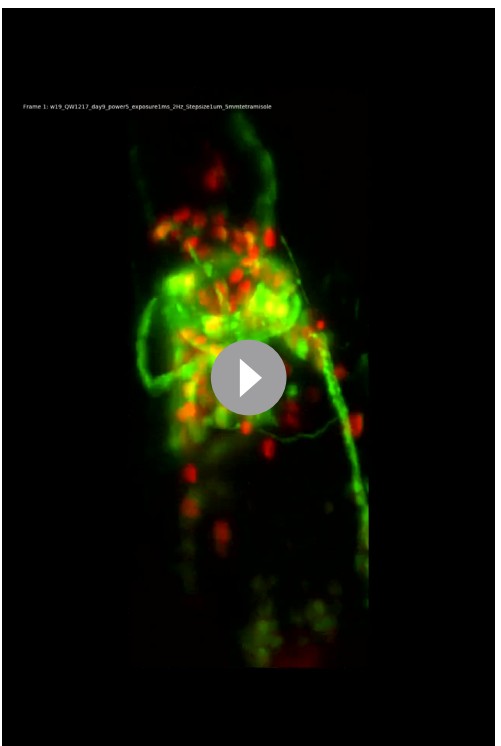

**Video 1.** Head region neural activity in day 1 adult *Caenorhabditis elegans*. Example videos of head region neural activity in five worms at day 1 of adulthood, displayed as two-dimensional projections. The entire 10 min of imaging per animal is sped up and played back over 30 s. Nuclear-localized RFP and cytoplasmic GCaMP fluorescences are displayed as red and green, respectively.

https://elifesciences.org/articles/72135/figures#video1

**Video 2.** Head region neural activity in day 9 adult *Caenorhabditis elegans*. Example videos of head region neural activity in five worms at day 9 of adulthood, displayed as two-dimensional projections. The entire 10 min of imaging per animal is sped up and played back over 30 s Nuclear-localized RFP and cytoplasmic GCaMP fluorescences are displayed as red and green, respectively.

https://elifesciences.org/articles/72135/figures#video2

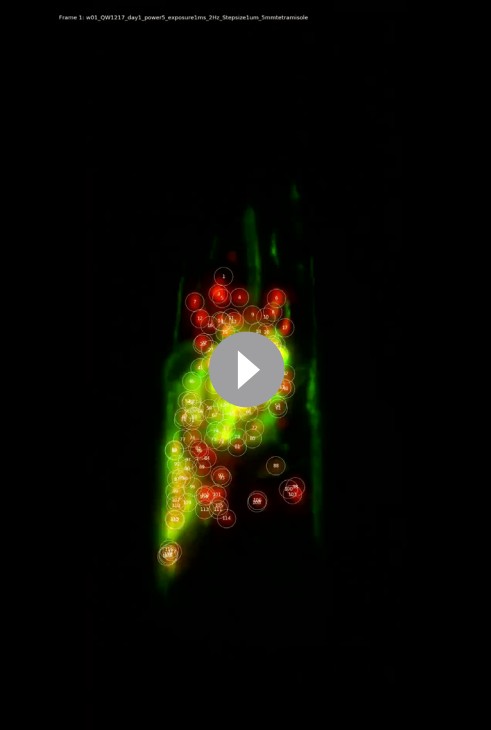

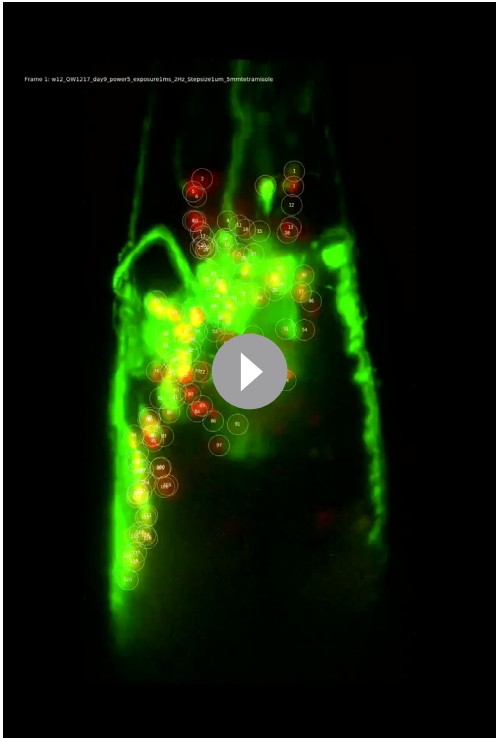

**Video 3.** Tracked head region neurons in day 1 adult *Caenorhabditis elegans.* Example videos of head region neural activity in five worms at day 1 of adulthood, displayed as two-dimensional projections. The entire 10 min of imaging per animal is sped up and played back over 30 s. Nuclear-localized RFP and cytoplasmic GCaMP fluorescences are displayed as red and green, respectively. White circles delineate the identified nuclear volumes of the tracked neurons.
https://elifesciences.org/articles/72135/figures#video3

**Video 4.** Tracked head region neurons in day 9 adult *Caenorhabditis elegans.* Example videos of head region neural activity in five worms at day 9 of adulthood, displayed as two-dimensional projections. The entire 10 min of imaging per animal is sped up and played back over 30 s. Nuclear-localized RFP and cytoplasmic GCaMP fluorescences are displayed as red and green, respectively. White circles delineate the identified nuclear volumes of the tracked neurons.
https://elifesciences.org/articles/72135/figures#video4

and methods) further displayed in *Videos 3 and 4*, respectively. Typical activity arrays from individual animals at day 1 and day 9 are shown in *Figure 2a*, while additional days 1, 3, 6, 9, and 12 examples are displayed in *Figure 2—figure supplement 2*. A decline in GCaMP fluorescence signal with age, due to factors such as decreases in GCaMP expression, GCaMP protein misfolding, or increased background fluorescence could potentially alter the measured neuronal signals. We, therefore, calculated the signal-to-noise ratio (SNR) of the GCaMP measurements across ages and found that there are no significant decreases (*Figure 2—figure supplement 1a*), despite an overall reduction in measured GCaMP intensities with age (*Figure 2—figure supplement 1c*). We further determined that there is no measurable change in tissue autofluorescence with age that might interfere with our GCaMP measurements (*Figure 2—figure supplement 1b*). As we and others have observed previously (*Kato et al., 2015*), young adult animals display a clear system-wide organization of neuronal dynamics, with large groups of neurons showing correlated activity (*Figure 2a*). The activity of such multi-neuron systems can be effectively presented using principal component analysis (PCA; *Bruno et al., 2017*), where the first three principal components (typically representing 70–90% of the variation in the *C. elegans* head region neurons) are visualized in a three-dimensional (3D) plot. As illustrated in *Figure 2c* and *Figure 2—figure supplement 2*, neuronal activity in young adult animals typically traces out a smooth trajectory in such PCA plots, which constitutes a well-defined activity manifold. Prior studies have demonstrated that regions along these traces represent particular behavioral states

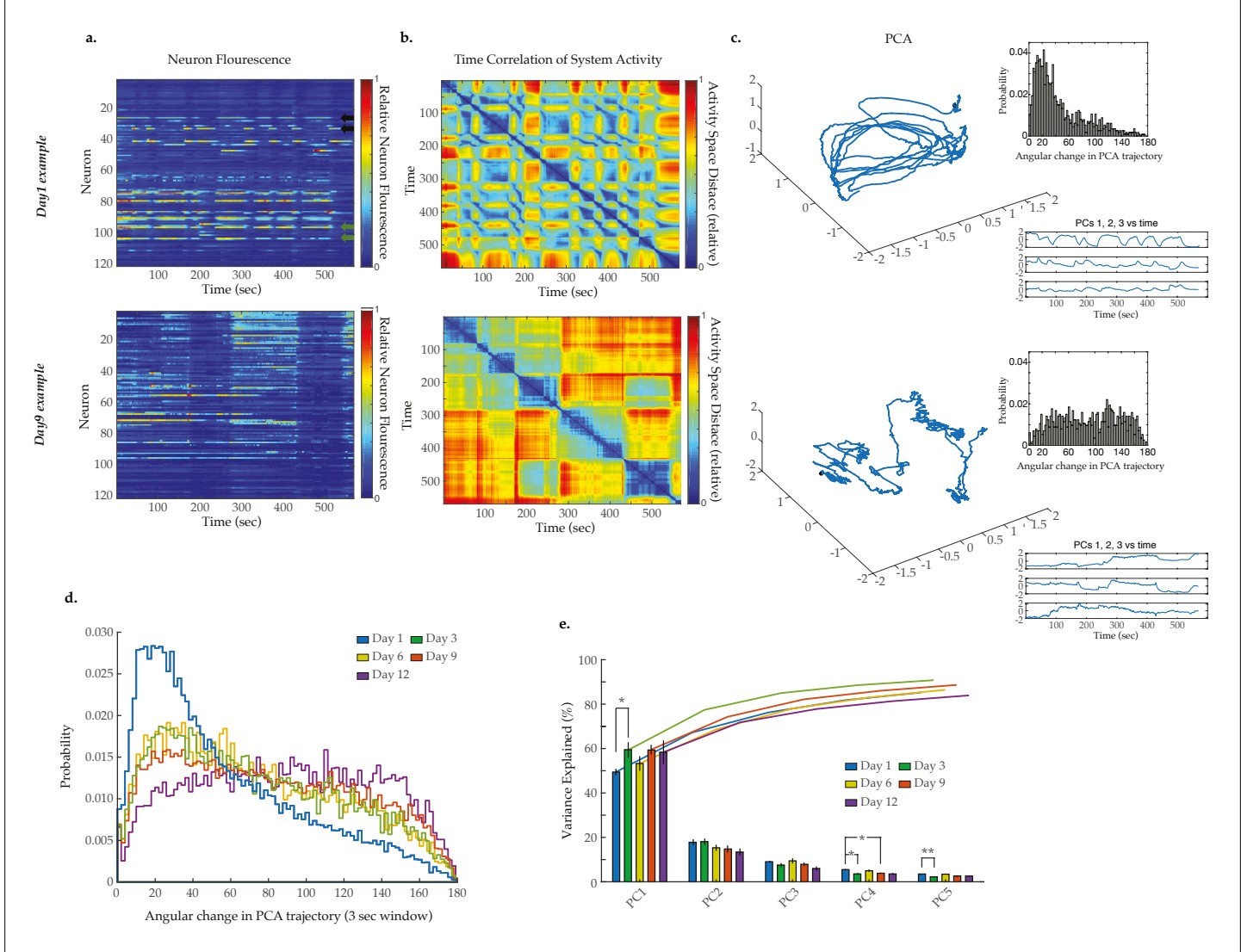

**Figure 2.** Breakdown of system dynamics and state organization with age. (**a**) Example traces of 120-neuron GCaMP fluorescence measurements, captured in the head region of a day 1 young adult and a day 9 senescent worm (blue and red represent scaled low and high fluorescence, respectively, following ΔF/F₀ normalization). Green and black arrows in the top panel highlight examples of highly correlated and anti-correlated neuron pairs, respectively. (**b**) Time correlation heatmaps derived of the fluorescence measurements in A. The relative distance in activity space is calculated between every pair of time points in the trial (see Materials and methods) indicated by color (blue indicates zero, red indicates maximal distance). (**c**) Three-dimensional plots displaying the trajectories of the first three principal components derived from fluorescence measurements in A. Top insets are probability histograms of the angular directional changes for those principal component trajectories (see Materials and methods). Bottom insets display the values of each of the first three principal components over time (**d**) Aggregate probability histograms of the angular directional changes in principal component trajectories for all animals measured at various ages. (**e**) Pareto plot with bars displaying the average variance explained by the first five principal components for each age group and lines displaying the cumulative total. (For D–E, n=20, 10, 10, 20, 10 animals for days 1, 3, 6, 9, and 12, respectively). Error bars denote standard error of the mean. *p<0.05; ** p<0.01; *** p<0.001, Sidak post hoc test.

The online version of this article includes the following source data and figure supplement(s) for figure 2:

**Source data 1.** Multi-neuron activity variance explained by principal components 1–5.

**Source data 2.** GCaMP signal-to-noise ratio with age.

**Source data 3.** Tissue autofluorescence among head region neurons.

**Figure supplement 1.** Signal-to-noise ratios of multi-neuronal GCaMP fluorescence measurements, tissue autofluorescence, and GCaMP intensities with age.

**Figure supplement 2.** Multi-neuron activity from individual worms with age.

of the animal (e.g. forward or backward movement or turning) that are linked by smooth transitions between states (*Kato et al., 2015*; *Awal et al., 2020*).

As *C. elegans* age, we observe a striking breakdown in the system organization. The lower panel of *Figure 2a* displays typical neuron activity measured in day 9 old animals. Apparent transitions between global states remain discernable by eye within the 120-neuron activity arrays. However, these states are less distinct and appear prolonged. This change is reflected in the accompanying PCA plot in the lower panel of *Figure 2c*. In contrast to the smooth transitions between repeating neuronal states observed in the young animal, this plot displays a much more erratic trajectory, with no apparent overall organization. The contrast between neuronal dynamics of young and old worms is further exemplified by the time correlation plots (*Figure 2b*, *Figure 2—figure supplement 2*). The neuronal activity at each time point is compared to that at every other time point by calculating the relative distance in activity space (see Materials and methods). Blue represents two time points with very similar neuronal activity patterns while red indicates significantly different neuronal activity patterns. In the young animal, there are a series of rapid transitions away from the reference activity pattern (yellow, red regions) and also transitions back to similar activity patterns (green, blue regions). In the older animal, these dynamics break down. Transitions between states are slower and more erratic. Additionally, the system does not readily return to previous activity patterns, indicating a breakdown in the recurrence of these system states.

To further quantify the breakdown in system-wide dynamics, we measured the smoothness of the PCA trajectories over time. At each time point, we calculated the absolute change in direction as the discrete time derivative of the tangential angle describing the trajectory. A smoothly curving trajectory will have small angular changes in direction over time, while a completely stochastic, randomized trajectory will have equally distributed changes in direction over the possible 180°. Example probability histograms of angular changes for individual worms are displayed in *Figure 2c* inset and *Figure 2—figure supplement 2*. Pooling all such measurements from all trials taken at a particular age, we generate an aggregate probability histogram of the angular directional changes of PCA trajectory for each age (*Figure 2d*). For young adult day 1 animals, this histogram is highly skewed toward smaller angles, reflecting the smooth characteristic of the trajectories and temporal continuity in the neuronal activity. As the animals age this bias is progressively lost, generating histograms with more equal distributions across all angular changes reflecting erratic randomly changing trajectories. This increased stochasticity of the PCA traces across the population of animals with age reflects the gradual loss of organization and temporal continuity in neuronal activity. This approach stands in contrast to directly comparing the system variance explained by each principal component (*Figure 2e*). In general, doing so does not reveal prominent age-dependent effects.

## Aged neurons exhibit higher-frequency activity and a specific loss of anti-correlativity

Within each ensemble of recorded neurons, it is possible to measure how individual neuron dynamics change with age. Power spectral density (PSD) analyses have been extensively used to quantitatively describe neuronal activity (*Buzsáki and Draguhn, 2004*; *Buzsáki and Watson, 2012*). We generated normalized PSDs for each of the 120 neurons imaged in each animal and averaged across all neurons measured within each age group (see Statistical Methods). Overlaying these average spectra revealed a progressive shift in relative power from low to higher frequencies with age (prominently from 0–0.05 to 0.10–0.15 Hz, respectively; *Figure 3a*). This shift is further illustrated by plotting each age group's cumulative power spectrum (*Figure 3a*, insert). For maximal visual clarity, average PSDs and cumulative power plots are also provided in *Figure 3—figure supplement 1* with standard error of the mean. To quantify this effect, we calculated the 40% spectral edge for each worm's PSD (i.e. the frequency below which 40% of the power resides, see Statistical methods). We observed a progressive increase in the average PSD 40% spectral edge with age (*Figure 3b*). The distribution of spectral edge measurements for all individual neurons within each age group is further provided in *Figure 3—figure supplement 2*. These distributions display an age-associated loss of neurons with low-frequency dynamics and an accompanying increase in neurons with high-frequency dynamics, in agreement with the mean PSD metrics in *Figure 3*. The shift in power spectra of the individual neurons to higher frequencies emphasizes the progressive loss of meaningful system-state dynamics observed above (*Figure 2a*, *Figure 2—figure supplement 2*). While the system-wide dynamics appear to slow with age, they also

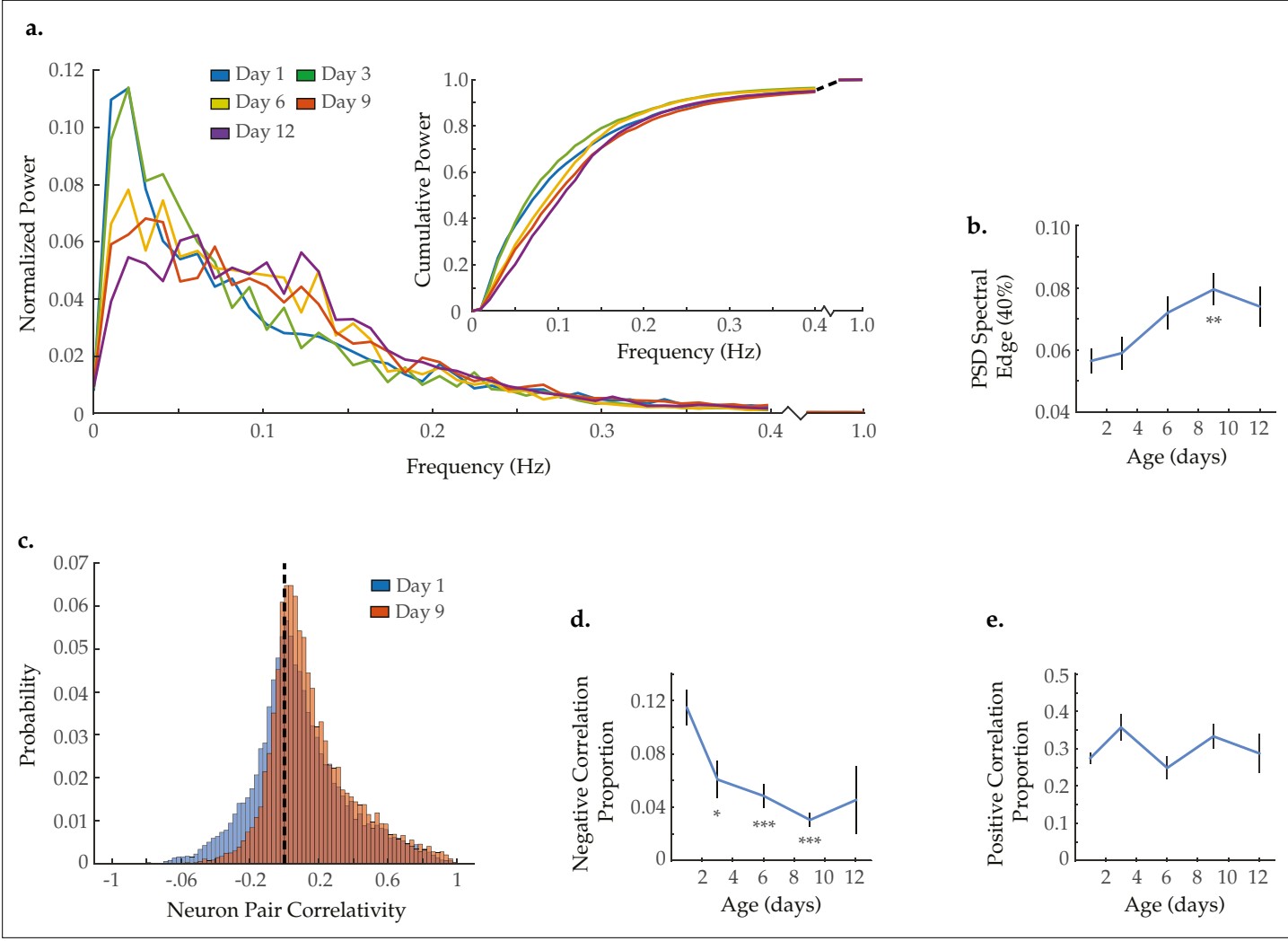

**Figure 3.** Neuron activity dynamics and correlation with age. (**a**) Mean power spectral densities (PSDs) derived from the fluorescence traces of individual neurons measured at various ages. Inset: cumulative PSD plots, display the proportion of power residing below a given frequency. Standard error of the mean for both PSD and cumulative power plots is displayed in *Figure 3—figure supplement 1*. (**b**) Mean 40% spectral edges across neuron PSDs at various ages, denoting the average frequency below which resides 40% of the total spectral power. For A and B, n=120 neurons per animal with 20, 10, 10, 20, and 10 animals for days 1, 3, 6, 9, and 12, respectively. (**c**) Aggregate probability histograms of neuron pair correlativity among the 40 most dynamically active neurons, in worms at days 1 and 9 of adulthood (see Materials and methods). (**d**) The negative correlation proportion, denoting the measured proportion of strongly negative neuron pair correlativity values (<−0.2). (**e**) The positive correlation proportion, denoting the measured proportion of strongly positive neuron pair correlativity values (>0.2). For C–E, n=40 neurons per animal with 20, 10, 10, 20, and 10 animals for days 1, 3, 6, 9, and 12, respectively. Error bars denote standard error of the mean. *p<0.05; ** p<0.01; *** p<0.001, Sidak post hoc test.

The online version of this article includes the following source data and figure supplement(s) for figure 3:

**Source data 1.** Multi-neuron frequency power distribution and activity correlativities with age.

**Source data 2.** Proportion of negatively correlated neuron pairs with 120 neurons analyzed per animal.

**Source data 3.** Multi-neuron frequency power distributions and proportion of negatively correlated neuron pairs in 10 and 20 min imaging trials.

**Figure supplement 1.** Power spectral density (PSD) and cumulative PSD plot error.

**Figure supplement 2.** Distribution of 40% spectral edges with age.

**Figure supplement 3.** Proportion of negatively correlated neuron pairs with age, when examining all 120 neurons captured per animal.

**Figure supplement 4.** Neuron pair correlativity and system dynamics in longer imaging trials.

become progressively less distinct and well defined such that they no longer measurably influence the activity dynamics of individual neurons.

The *C. elegans* nervous system possesses a subset of strongly connected neurons displaying highly correlated or anti-correlated activities (*Cook et al., 2019*). Such connections are essential for the effective organization and function of a nervous system. Inspection of the activity array from the young adult animal in *Figure 2a* reveals both highly correlated pairs (green arrows) as well as highly anti-correlated pairs (black arrows). To measure changes in neuronal connectivity with age, we calculated the signal correlativity between all possible neuron pairs among the 40 most dynamically active neurons in each animal. We then pooled all correlation measurements at each age to generate an aggregate probability histogram of neuron-neuron correlations (*Figure 3c*). We observe a reduction in the proportion of strongly anti-correlated neuron pairs with age. In contrast, the proportion of highly correlated neuron pairs remains relatively stable. To quantify this effect, we calculated the proportion of neuron pairs demonstrating strong negative or positive correlativity (i.e. the negative and positive correlation proportion, respectively) for each worm and averaged by age (see Statistical methods). We observe a significant reduction in the negative correlation proportion occurring as early as day 3 of adulthood and further progressing with age (*Figure 3d*). However, no change is observed in the positive correlation proportion (*Figure 3e*). These calculations were also performed with all 120 neurons captured per animal (*Figure 3—figure supplement 3*) as well as with longer imaging trials (*Figure 3—figure supplement 4*), yielding consistent results. The loss of neuronal pairs that are negatively correlated suggests a specific breakdown in inhibitory signaling with age. The accompanying persistence of positively correlated neuronal pairs (brought about by excitatory signaling) points to a resultant shift in the excitatory/inhibitory balance of the nervous system.

## Aged animals exhibit periods of global neural quiescence

Interestingly, visual inspection of neuron activity, displayed in heatmap arrays in *Figure 2a* and *Figure 2—figure supplement 2*, as well as composite activity plots in *Figure 4a*, suggests that aged animals enter periods of global neural quiescence, in which a large majority of neurons maintain low activity. Such quiescent periods are not evident in young adults. To quantify this, we determined how many neurons in each animal were quiescent at each time point (i.e. displayed low fluorescence level below a set cutoff value, see Materials and methods). We pooled the proportion of quiescent neurons at each time point across all trials to generate a probability distribution of neuron quiescence at each age (*Figure 4b*). Comparing the distributions of day 1 versus day 9 worms, we observe a shared peak at 40% neural quiescence and also a peak unique to the aged condition centered around 80% neural quiescence. These two peaks are consistent, respectively, with an aroused state and quiescent sleep-like state. Similar to our results, a prior study quantified the proportion of neurons inactive during aroused and sleep-like behavior as 60 and 90%, respectively (*Nichols et al., 2017*). For our study, we classified global neural quiescence as time points in which at least 70% of neurons in an animal were quiescent (dashed line in *Figure 4b*). Examples of individual trials are shown in *Figure 4a*; red dots beneath the traces denote time points that meet the criteria for global quiescence. For each age group, we calculated the average proportion of time that animals were in global neural quiescence (*Figure 4c*). While day 1 worms display global quiescence 1% of the time, on average, day 9 worms do so 9% of the time. These results demonstrate that aged worms enter periods of global neural quiescence during our 10 min imaging protocol, potentially reflective of sleep-like behavior, while young adult worms largely do not.

To determine if bouts of global neural quiescence account for the age-associated shifts in spectral power, neuron pair correlativity, and system dynamics, we compared day 9 activity arrays that contain periods of quiescence to those that do not in two distinct data subsets (see Materials and methods; *Figure 4—figure supplement 1*). We find that in the presence or absence of bouts of global quiescence, a dramatic reduction in negative neuron pair correlativity is still observed in day 9 worms, as compared to young adults. We find that the positive neuron pair correlativity increases in the quiescence day 9 subset compared to the non-quiescence data set, as would be expected with concerted transitions into and out of quiescence (see Materials and methods). However, this effect is undetectable for the complete day 9 data set (*Figure 3e*) in which global quiescence is less prevalent (~9% time in quiescence for the complete day 9 data set compared to ~17% time in the quiescence data subset). Furthermore, we find that the shift from low- to high-frequency spectral power

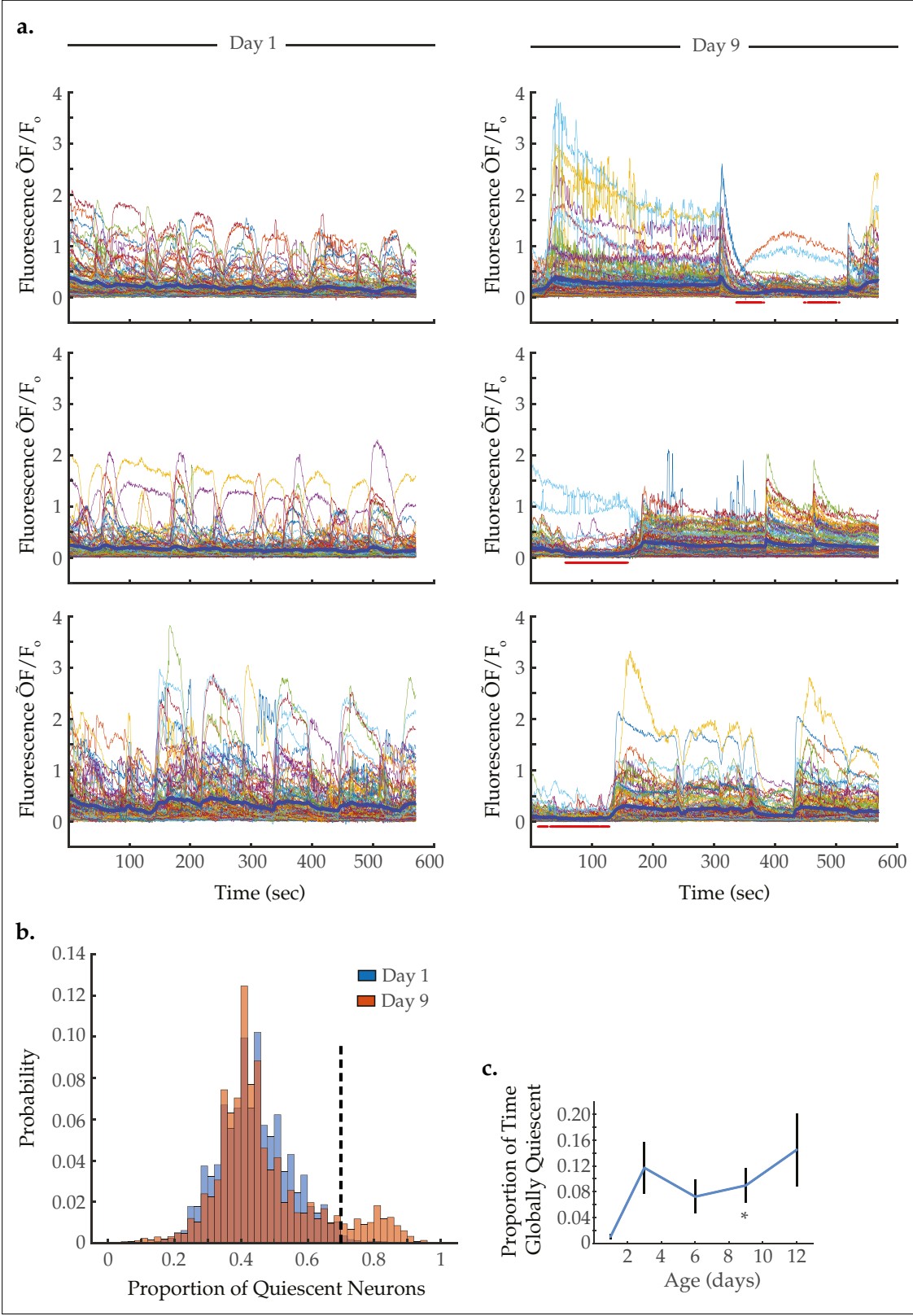

**Figure 4.** Global neural quiescence with age. (**a**) Example traces of 120-neuron GCaMP fluorescence (ΔF/F₀) measurements, captured in the head region of three day 1 young adults and three day 9 senescent worms. The thick blue line represents the average normalized GCaMP fluorescence at each time point. The red bars beneath the day 9 traces demarcate time spans identified as exhibiting global neural quiescence (see Materials and methods). (**b**) Histograms displaying the probability distribution of the proportion of neurons quiescent during a given time point, in worms at days 1

*Figure 4 continued on next page*

*Figure 4 continued*

and 9 of adulthood (see Materials and methods). (**c**) The proportion of time the head region neurons exhibit global quiescence, in worms at days 1, 3, 6, 9, and 12 of adulthood (see Materials and methods). For B–C, n=120 neurons per animal with 20, 10, 10, 20, and 10 animals for days 1, 3, 6, 9, and 12, respectively. Error bars denote standard error of the mean. *p<0.05; ** p<0.01; *** p<0.001, Sidak post hoc test.

The online version of this article includes the following source data and figure supplement(s) for figure 4:

**Source data 1.** Extent of global neural quiescence with age.

**Source data 2.** Impact of global neural quiescence on multi-neuron frequency power distributions and activity correlativities.

**Figure supplement 1.** Impact of global neural quiescence on neural activity spectral power, correlation, and system dynamics in aged *Caenorhabditis elegans*.

(*Figure 4—figure supplement 1c*) is still observed in day 9 worms with and without global quiescence. While the age-associated increase in 40% spectral edge is no longer statistically significant with the smaller data subsets, the trend remains (*Figure 4—figure supplement 1d*). Finally, the breakdown in system temporal continuity remains essentially unchanged in quiescent versus non-quiescent data sets (*Figure 4—figure supplement 1e*). Thus, while increased bouts of global neural quiescence are an intriguing aspect of aging, it does not appear to be sufficient to explain the additional changes in neuron dynamics that we observe. *Figure 4—figure supplement 1f*, displays the average proportions of time in global neural quiescence for each condition examined in this study.

## Age-associated changes to neuronal dynamics are recapitulated in young *unc-2* gain-of-function mutant animals

To investigate the molecular mechanisms that could underlie age-related changes in network behavior, we considered candidate genes known to regulate excitatory/inhibitory balance. Mutations in the *unc-2*/CaV2α gene have been shown to alter the excitatory/inhibitory signaling balance in the *C. elegans* nervous system (*Huang et al., 2019*). *unc-2* encodes the *C. elegans* ortholog of the pore-forming alpha-1A subunit of the voltage-dependent P/Q-type calcium channel (CACNA1A). Animals with the gain-of-function (gf) mutation *unc-2(zf35)* show an increased rate of spontaneous reversals as young adults (*Figure 5a*), similar to that of aged wild-type animals. Multi-neuron imaging in *unc-2(gf)* animals, revealed striking similarities between the neuronal dynamics of day 1 *unc-2(gf)* animals and those of older (day 9) wild-type animals. Typical activity arrays from individual day 1 and 9 *unc-2(gf)* animals are shown in *Figure 5—figure supplement 1*. As early as day 1, the average power spectra of this mutant display a shift from low- to high-frequency power similar to that seen in wild-type aging (*Figure 5b*). The 40% spectral edge is significantly increased at day 1 in *unc-2(gf)* animals, relative to wild-type (*Figure 5c*). The distribution of neuron pair correlations was similarly affected by the mutation, as the age-related changes were recapitulated at a younger age in *unc-2(gf)* animals (*Figure 5d*). The negative correlation proportion is significantly decreased at day 1 in *unc-2(gf)* animals, relative to wild-type (*Figure 5e*), while the positive correlation proportion is unchanged across ages and genotypes (*Figure 5—figure supplement 2*). Likewise, the temporal continuity of system activity in *unc-2(gf)* animals breaks down at an early age, with the probability histogram of angular directional changes of PCA trajectory from day 1 *unc-2(gf)* worms closely matching that of day 9 wild-type animals (*Figure 5f*). These experiments demonstrate that UNC-2/CaV2α gf results in increased crawling reversal rate and neuronal hyperactivity, a loss of inhibitory signaling, and disruption of system organization in young animals, all of which closely match the effects observed with aging.

We next examined the canonical loss-of-function (lf) *unc-2(e55)* mutant strain recently shown to decrease the neuronal excitatory/inhibitory balance in *C. elegans* (*Huang et al., 2019*). Young adult *unc-2(lf)* worms show lethargic crawling behavior and a decreased spontaneous reversal rate throughout their lifespan (*Figure 5a*). We performed multi-neuronal imaging on the *unc-2(lf)* animals at day 1 and day 9 of adulthood. Typical activity arrays from individual day 1 and 9 *unc-2(lf)* animals are shown in *Figure 5—figure supplement 1*. We find that the PSD of *unc-2(lf)* animals shifts to higher frequencies to a far lesser extent with age (*Figure 5b and c*). Likewise, the 40% spectral edge is trending lower at day 9, relative to wild type (p=0.05; *Figure 5c*). However, age-related changes to the neuron pair correlativity appear essentially unaffected (*Figure 5d*), as the negative correlation proportion at day 9 remains comparable to wild type (*Figure 5e*). Moreover, the breakdown in temporal continuity of system activity appears largely unchanged with only a slight shift in the

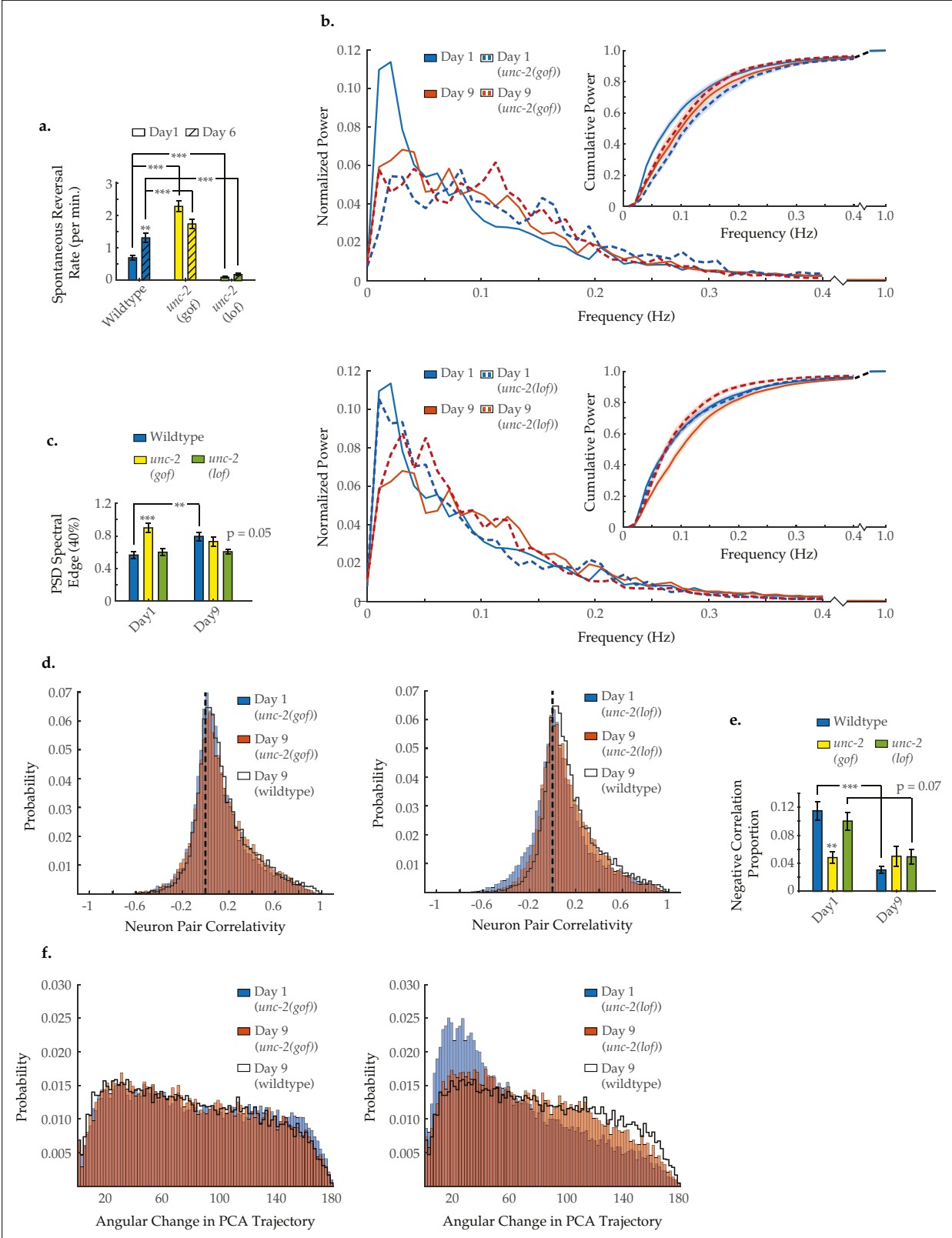

**Figure 5.** Neuron activity dynamics of *unc-2* mutant animals with age. (**a**) Spontaneous reversal rates of QW1217 strain worms on a wild-type, *unc-2(e55)* loss-of-function (lf), and *unc-2(zf35)* gain-of-function (gf) background, on days 1 and 6 of adulthood. On days 1 and 6, n=28 and 26 for wild-type worms, 26 and 26 for *unc-2(lf)* worms, and 25 and 22 for *unc-2(gf)* worms, respectively. (**b**) Mean power spectral densities (PSDs) of the neuron GCaMP fluorescence in wild-type, *unc-2(lf)*, and *unc-2(gf)* worms on days 1 and 9 of adulthood. Insert: cumulative PSD plots. Standard error of the mean for both

*Figure 5 continued on next page*

*Figure 5 continued*

PSD and cumulative power plots is displayed in *Figure 3—figure supplement 1*. (**c**) Mean 40% spectral edges of the individual PSDs used to generate graphs in B. For B and C, n=120 neurons per animal with 17–20 animals per condition. (**d**) Aggregate probability histograms, and (**e**) the negative correlation proportion, of neuron pair correlativity among the 40 most dynamically active neurons, in wild-type, *unc-2(lf)*, and *unc-2(gf)* worms on days 1 and 9 of adulthood. For D and E, n=40 neurons per animal with 17–20 animals per condition. (**f**) Aggregate probability histograms of the angular directional changes in principal component analysis (PCA) trajectories for wild-type, *unc-2(lf)*, and *unc-2(gf)* worms on days 1 and 9 of adulthood. Each PCA trajectory was generated with 120 neurons per animal with 17–20 animals per condition. Wild-type data displayed here are the same as displayed in *Figures 2 and 3*. Error bars denote standard error of the mean. *p<0.05; ** p<0.01; *** p<0.001, Sidak post hoc test.

The online version of this article includes the following source data and figure supplement(s) for figure 5:

**Source data 1.** Reversal behavior in *unc-2* mutant worms.

**Source data 2.** Multi-neuron frequency power distributions and proportion of negatively correlated neuron pairs in *unc-2* mutants.

**Source data 3.** Proportions of positively correlated neuron pairs across genotypes and drug treatments.

**Figure supplement 1.** Multi-neuron activity from individual worms with age and genotype.

**Figure supplement 2.** Proportion of positively correlated neuron pairs with age.

distribution of PCA angular directional changes at day 9 compared to wild type (*Figure 5F*). Thus, UNC-2/CaV2α(lf) appears to preserve many, but not all aspects, of neuronal signaling that are typically altered with age.

## Age-associated changes to neuronal dynamics are partially rescued in aged *ced-4* loss-of-function mutants

During development, UNC-2-mediated calcium signaling has been directly linked to the removal of presynaptic GABAergic complexes in *C. elegans* motor neurons (*Miller-Fleming et al., 2016*). This signaling triggers CED-3 caspase activity to degrade these presynaptic domains and is blocked by an lf mutation in its upstream regulator CED-4/Apaf-1 (*Miller-Fleming et al., 2016*). To test if a similar process could be at work during aging, we tested an lf *ced-4(n1162)* mutant strain in our imaging assays. Typical activity arrays from individual day 1 and 9 *ced-4* animals are shown in *Figure 5—figure supplement 1*. In these animals, we did not observe the characteristic shift from low to high frequency in the neuronal activity PSD, resulting in no significant change in the 40% spectral edge with age (*Figure 6a and b*). Likewise, the loss of strongly anti-correlated neuron pairs with age was diminished (*Figure 6c*). The negative correlation proportion in ced-4 worms is not significantly changed with age, when comparing days 1 and 9, and is significantly higher on day 9, as compared to wild-type day 9 (*Figure 6d*). Similar to the *unc-2(lf)* mutant, the age-dependent breakdown in system-wide temporal continuity was largely unchanged in *ced-4* animals, with only a very mild shift in the distribution of PCA angular directional changes at day 9 compared to wild type (*Figure 6e*). These results show similarities to that of *unc-2(lf)*, suggesting that they may indeed contribute to the same cellular processes affecting neuronal aging.

## Agonizing GABA_A receptors partially restores neuronal dynamics in aged worms

To investigate if a breakdown in inhibitory GABA signaling plays an important role in the age-dependent changes to neural dynamics, we applied the GABA_A agonist muscimol to young adult and aged worms. Typical activity arrays from individual day 1 and day 9 animals treated with muscimol are shown in *Figure 5—figure supplement 1*. We found that acute muscimol application reshapes the neuronal activity PSD in a manner opposite to the effects of aging, causing a shift from high- to low-frequency power in both young adult and aged animals and significantly lowering the 40% spectral edge in day 9 animals (*Figure 7a and b*). Unsurprisingly, there is no effect on the age-dependent loss of negative correlation between neurons in muscimol-treated animals (*Figure 7c and d*), as this exogenous ligand agonizes GABA_A receptors across the nervous system irrespective of neuronal connectivity. In addition, muscimol treatment moderately rescues system organization in the aged animal, as PCA analysis reveals better temporal continuity in neuronal dynamics (i.e. the distribution of PCA angular directional changes retains more bias toward smaller angles in day 9 animals; *Figure 7e*). Taken as a whole, these results demonstrate that pharmacological stimulation of GABA_A signaling can counteract many of the age-dependent effects we observe in *C. elegans* neuronal dynamics.

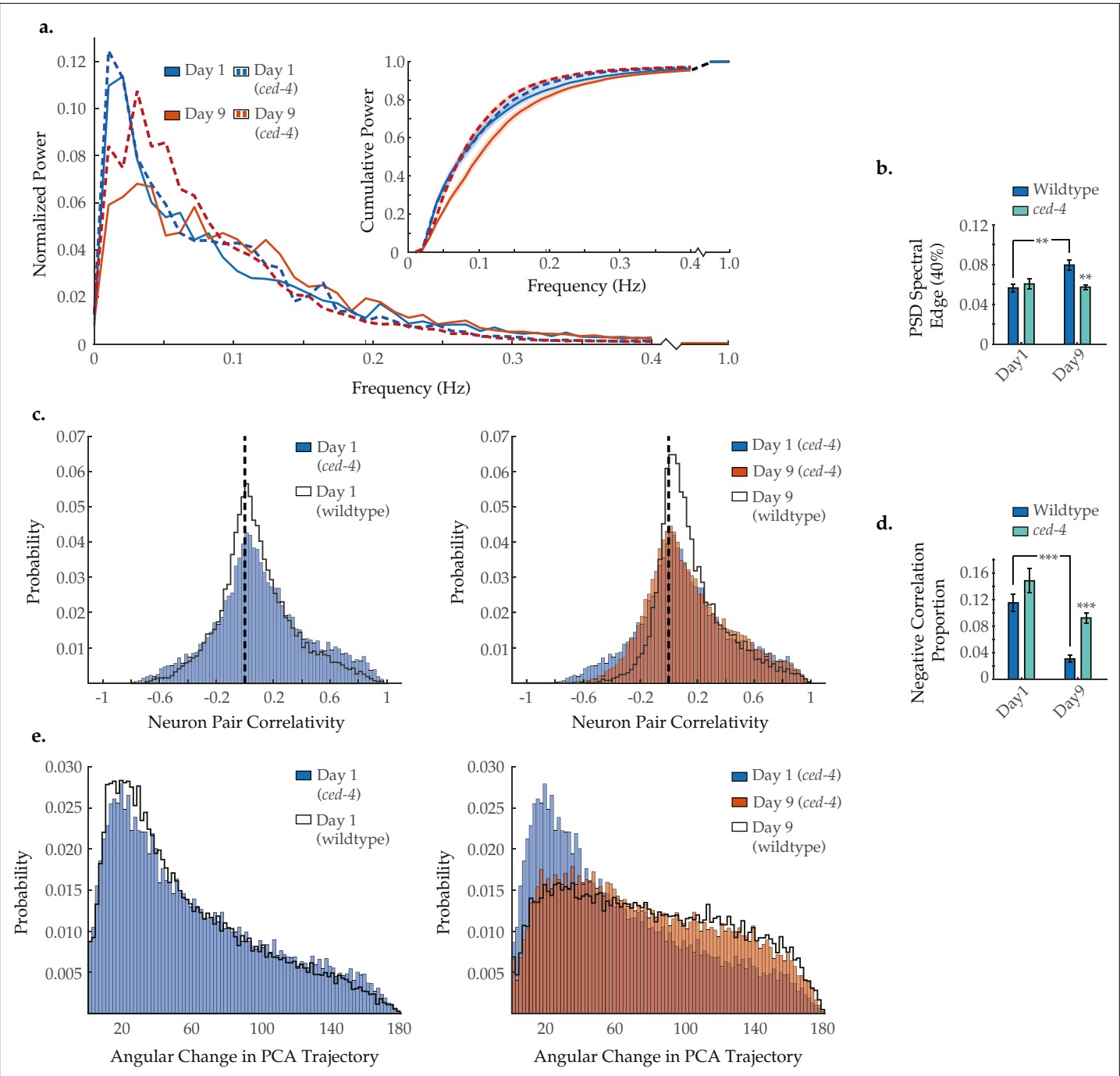

**Figure 6.** C*ed-4* mutation diminishes age-associated changes in neuronal dynamics. (**a**) Mean power spectral densities (PSDs) of the neuron GCaMP fluorescence traces in wild-type and *ced-4(n1162)* loss-of-function worms on days 1 and 9 of adulthood. Insert: cumulative PSD plots. Standard error of the mean for both PSD and cumulative power plots is displayed in *Figure 3—figure supplement 1*. (**b**) Mean 40% spectral edges of the individual PSDs used to generate graphs in A. For A and B, n=120 neurons per animal with 18–20 animals per condition. (**c**) Aggregate probability histograms, and (**d**) the negative correlation proportion, of neuron pair correlativity among the 40 most dynamically active neurons, in wild-type and *ced-4* worms on days 1 and 9 of adulthood. For C and D, n=40 neurons per animal with 18–20 animals per condition. (**e**) Aggregate probability histograms of the angular directional changes of principal component analysis (PCA) trajectories for wild-type and *ced-4* worms on days 1 and 9 of adulthood. Each PCA trajectory was generated with 120 neurons per animal with 18–20 animals per condition. Wild-type data displayed here are the same as displayed in *Figures 2 and 3*. Error bars denote standard error of the mean. *p<0.05; ** p<0.01; *** p<0.001, Sidak post hoc test.

The online version of this article includes the following source data for figure 6:

**Source data 1.** Multi-neuron frequency power distributions and proportion of negatively correlated neuron pairs in a *ced-4* mutant worm.

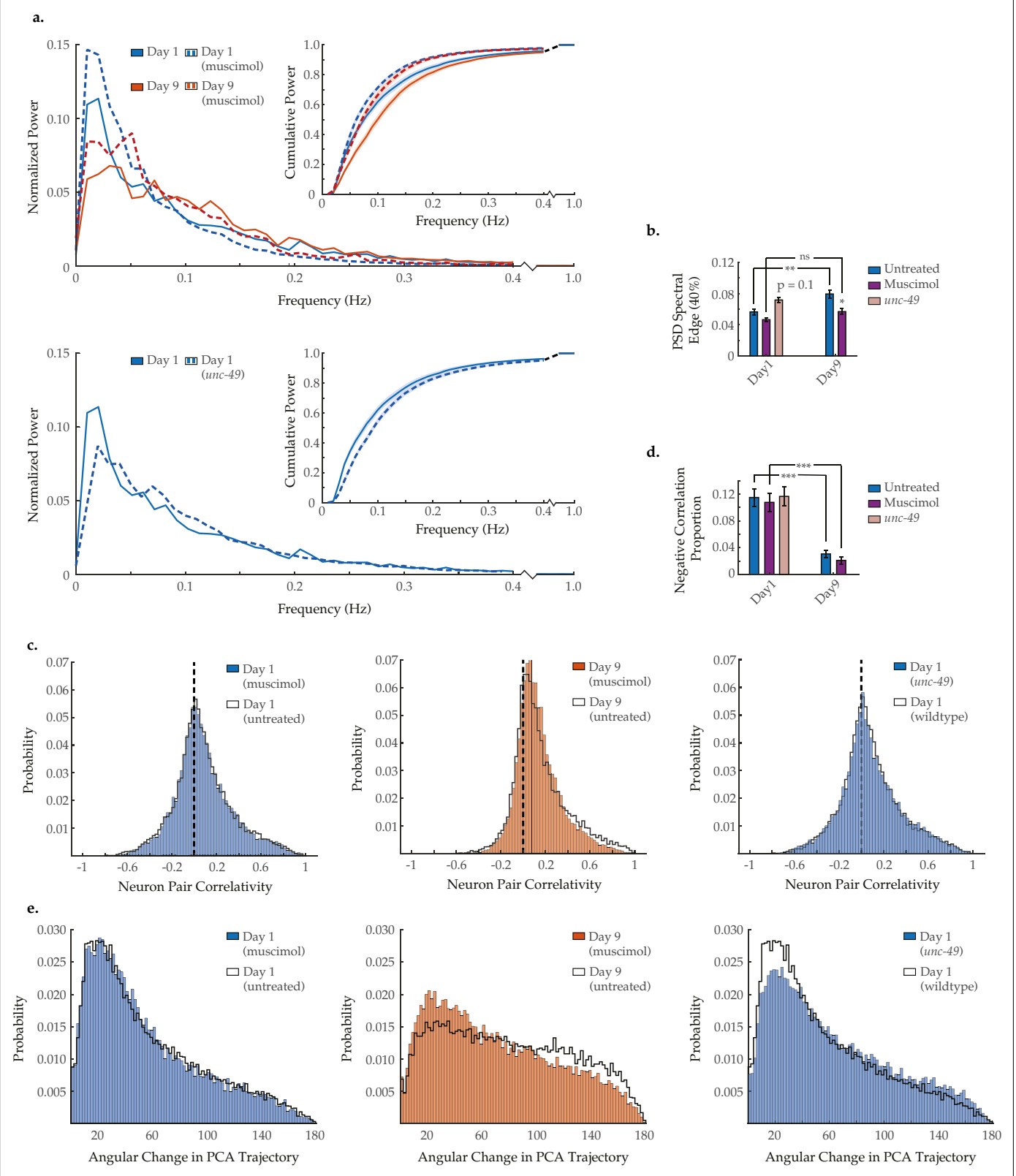

**Figure 7.** Modulating GABA_A signaling alters several age-associated changes in neuron dynamics. (**a**) Mean power spectral densities (PSDs) of the neuron GCaMP fluorescence traces in wild-type untreated and muscimol-treated animals on days 1 and 9 of adulthood, and also *unc-49(e407)* animals on day 1 of adulthood. Insert: cumulative PSD plots. Standard error of the mean for both PSD and cumulative power plots is displayed in *Figure 3— figure supplement 1*. (**b**) Mean 40% spectral edges of the individual PSDs used to generate graphs in A. For A and B, n=120 neurons per animal with

*Figure 7 continued on next page*

*Figure 7 continued*

17–20 animals per condition. (**c**) Aggregate probability histograms, and (**d**) the negative correlation proportion, of neuron pair correlativity among the 40 most dynamically active neurons, in wild-type untreated and muscimol-treated worms on days 1 and 9 of adulthood, and also *unc-49(e407)* animals on day 1 of adulthood. For C and D, n=40 neurons per animal with 17–20 animals per condition. (**e**) Aggregate probability histograms of the angular directional changes of principal component analysis (PCA) trajectories in wild-type untreated and muscimol-treated worms on days 1 and 9 of adulthood, and also *unc-49(e407)* animals on day 1 of adulthood. Each PCA trajectory was generated with 120 neurons per animal with 17–20 animals per condition. Untreated data displayed here are the same as displayed in *Figures 2 and 3*. Error bars denote standard error of the mean. ns, not significant; p>0.05; *p<0.05; ** p<0.01; *** p<0.001, Sidak post hoc test.

The online version of this article includes the following source data for figure 7:

**Source data 1.** Multi-neuron frequency power distributions and proportion of negatively correlated neuron pairs in muscimol-treated animals and an *unc-49* mutant worm.

We further examined the *unc-49(e407)* mutant strain, which carries a premature stop codon in the gene encoding GABA$_A$ receptor UNC-49. Typical activity arrays from individual *unc-49* day 1 animals are shown in *Figure 5—figure supplement 1*. In young adults, we find that the neuronal activity PSD is reshaped in a manner reminiscent of aging, with a shift from low- to high-frequency power, yielding a 40% spectral edge that trends higher than that of wild-type animals (p=0.1; *Figure 7a and b*). Interestingly, we observe no decrease in the baseline prevalence of negative correlation between neurons in young adults (*Figure 7c and d*). However, we do observe a breakdown in system organization, with PCA analysis revealing poorer temporal continuity in neuronal dynamics (*Figure 7e*). While the effects in *unc-49(e407)* worms are mild, possibly due to developmental compensation, they are consistent with, that is, opposite to, those we describe in muscimol-treated animals. This lends further credence to the observed effects of muscimol on neuronal dynamics being due to GABA$_A$ agonization, as opposed to unrelated drug effects.

## Long-lived *daf-2* mutation alters neuronal dynamics with age

To further assess the effects of aging on neuronal dynamics, we examined a genetic strain with extended life- and healthspans. *C. elegans* harboring an lf mutation within the insulin/ IGF-1 receptor ortholog *daf-2* displays robust longevity (*Kenyon et al., 1993*). *daf-2* reversal behavior is curious, as young adults have been shown to display a high reversal rate that is maintained with age (*Figure 8—figure supplement 1a*; *Podshivalova et al., 2017*). Imaging the AVA neuron individually, we find that AVA activation and inactivation dynamics are fully preserved at day 9 (*Figure 8—figure supplement 1b–d*), but the interneuron's duty ratio is still altered with age, similar to wild type (*Figure 8—figure supplement 1e*). Performing multi-neuronal imaging, we find that *daf-2* mutation appears to prevent or delay some but not all characteristics of normal aging. Typical activity arrays from individual *daf-2* day 1 and day 9 animals are shown in *Figure 5—figure supplement 1*. The shift in neuronal activity toward higher frequencies (*Figure 8a*) appears to be diminished, with the 40% spectral edge of day 9 *daf-2* worms showing no significant difference from day 1 *daf-2* and trending lower than day 9 wild-type animals (p=0.12; *Figure 8b*). Likewise, age-dependent breakdown in system-wide temporal continuity was somewhat diminished in *daf-2* animals, with the distribution of PCA angular directional changes retaining more bias toward smaller angles (*Figure 8e*). However, the loss of strongly anti-correlated neuron pairs with age was unaffected by *daf-2* mutation, with the negative correlation proportion changing with age comparably to wild type (*Figure 8c and d*).

## Discussion

Despite the simplicity of the *C. elegans* nervous system, it has the capacity to mediate a variety of behaviors, ranging from spontaneous crawling and sensory response to complex chemotaxis and associative learning, all of which exhibit age-associated decline (*Glenn et al., 2004*; *Kauffman et al., 2010*; *Churgin et al., 2017*). Such behavioral changes are inherently interconnected. For example, a simple change in spontaneous reversal rate could alter the animal's roaming behavior or its ability to effectively chemotax. Moreover, the animal is subject to the progressive deterioration of muscle function and locomotion. Taken as a whole, it is difficult to separate out the critical aspects of neuronal decline from the animal's physical deterioration and breakdown of complex behaviors. To this end, recent studies have begun to gain insight into the alterations of the nervous system that underlie

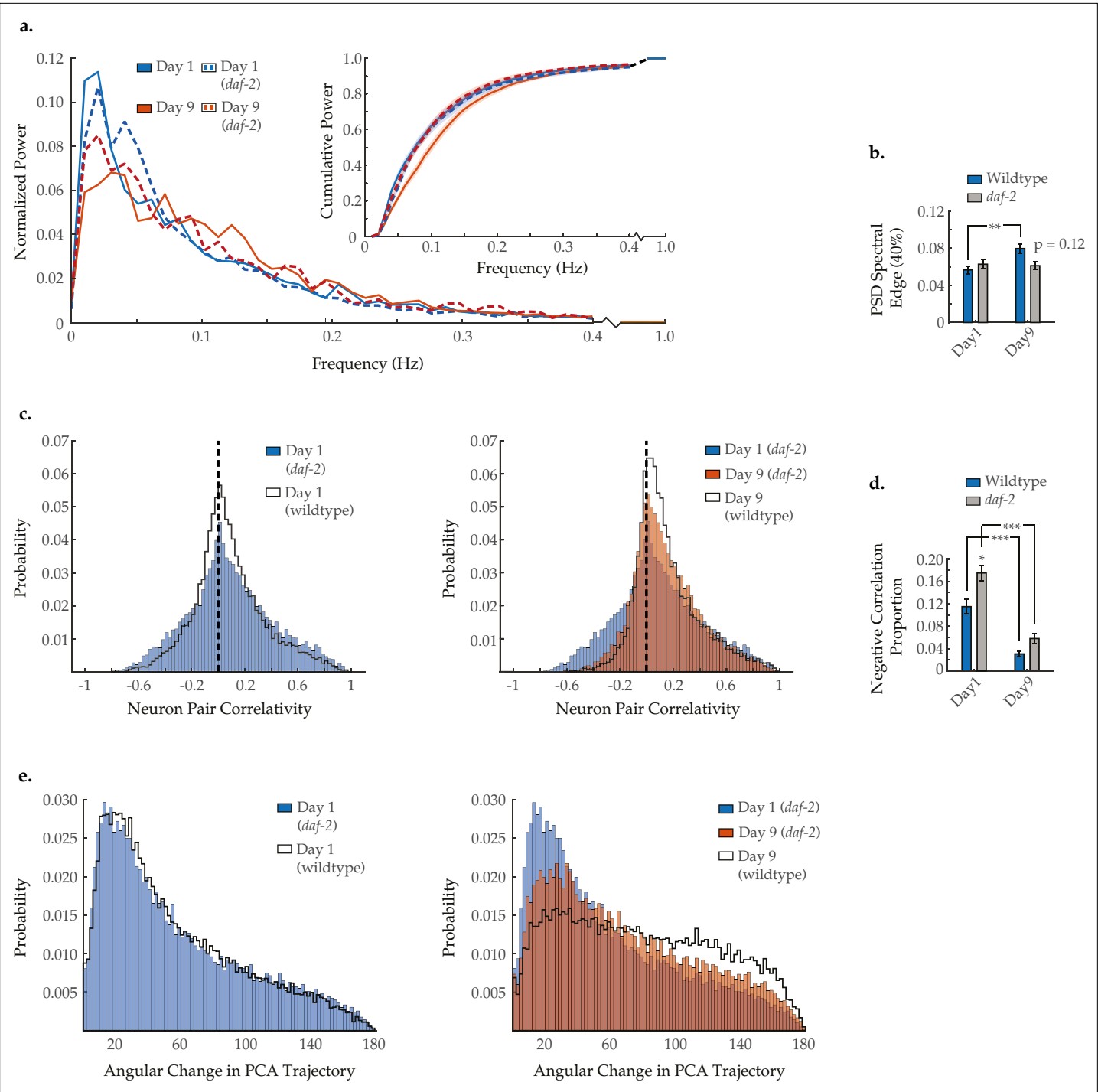

**Figure 8.** Effects of long-lived d*af-2* mutation on age-associated changes in neuronal dynamics. (**a**) Mean power spectral densities (PSDs) of the neuron GCaMP fluorescence traces, in wild-type and *daf-2(e1370)* loss-of-function worms on days 1 and 9 of adulthood. Insert: cumulative PSD plots. Standard error of the mean for both PSD and cumulative power plots is displayed in *Figure 3—figure supplement 1*. (**b**) Mean 40% spectral edges of the individual PSDs used to generate graphs in A. For A and B, n=120 neurons per animal with 18–20 animals per condition. (**c**) Aggregate probability histograms, and (**d**) the negative correlation proportion of neuron pair correlativity among the 40 most dynamically active head region neurons, in wild-type and *daf-2* worms on days 1 and 9 of adulthood. For C and D, n=40 neurons per animal with 18–20 animals per condition. (**e**) Aggregate probability histograms of the angular directional changes of principal component analysis (PCA) trajectories for wild-type and *daf-2* worms on days 1 and 9 of adulthood. Each PCA trajectory was generated with 120 neurons per animal with 18–20 animals per condition. Wild-type data displayed here are the same as displayed in *Figures 2 and 3*. Error bars denote standard error of the mean. *p<0.05; ** p<0.01; *** p<0.001, Sidak post hoc test.

The online version of this article includes the following source data and figure supplement(s) for figure 8:

*Figure 8 continued on next page*

*Figure 8 continued*

**Source data 1.** Multi-neuron frequency power distributions and proportion of negatively correlated neuron pairs in a *daf-2* mutant worm.

**Source data 2.** Reversal behavior in *daf-2* mutant worms.

**Source data 3.** AVA interneuron activity state transition dynamics in a *daf-2* mutant worm.

**Source data 4.** AVA interneuron calcium transient duty ratio in a *daf-2* mutant worm.

**Figure supplement 1.** AVA activity dynamics in *daf-2* mutant animals with age.

this functional decline with age, including changes to neurite ultrastructure (*Pan et al., 2011*; *Tank et al., 2011*; *Toth et al., 2012*), synaptic transmission (*Liu et al., 2013*; *Mulcahy et al., 2013*), and basal activity and excitability of specific neurons (*Chokshi et al., 2010*; *Liu et al., 2013*; *Leinwand et al., 2015*; *Zullo et al., 2019*; *Huang et al., 2020*). In this study, we expand these efforts to include advanced multi-neuron fluorescence imaging that captures the activity dynamics across large swaths of the nervous system at single-cell resolution. Measuring global neural activity in aged *C. elegans* revealed novel age-associated changes in neuronal signaling and activity that contribute to the decline of system-wide dynamics and function.

With age, we observe a striking breakdown in system-wide organization and temporal continuity that accompanies a shift in neuronal dynamics toward higher frequencies. System-wide analysis reveals a loss of well-defined and recurrent system states that are observed in young adult animals, visualized using both principal component trajectories (*Figure 2c*, *Figure 2—figure supplement 2*) and time correlation heatmaps (*Figure 2b*, *Figure 2—figure supplement 2*). While young worms display rapid transitions between repeated states, old worms appear to dwell for longer periods of time in poorly defined states, transition slowly between states, and do not efficiently return to prior states. These age-associated changes in global system dynamics are congruent with the prolonged calcium transients and slower state transitions we observe in the AVA command interneuron (*Figure 1f–i*). Likewise, the power spectral analysis of individual neuron activity (*Figure 3a and d*) reveals a progressive loss of low-frequency power associated with distinct state transitions and a corresponding increase at higher frequencies. The principal component trajectories of older animals display progressively more random, erratic changes in direction over short time periods, as measured by the distribution of angular changes (*Figure 2d*). These observations reflect a system that has lost temporal continuity and no longer adheres to a well-defined activity manifold. Interestingly, the most dramatic changes in system organization occur as early as day 3 of adulthood, suggesting that breakdown of system-wide state dynamics begins remarkably early in the aging process. Taken together, our findings describe a system in which age-associated changes result in randomization of system dynamics, accompanied by an increase in high-frequency activity at the neuronal level, and ultimately a breakdown of well-defined system states.

With advancing age, we also measure a progressive reduction in anti-correlated neuronal activity (*Figure 3c*), indicative of diminished inhibitory signaling. However, this is not accompanied by a loss of positive correlativity, reflecting a maintenance of excitatory signaling. This specific loss of inhibitory signaling with age shifts the system-wide excitatory/inhibitory balance toward excitation (*Figure 3d and e*). Again, these effects begin as early as day 3 of adulthood. Maintaining an appropriate excitatory/inhibitory balance is essential to the health and function of the nervous system. Increased excitability has been associated with aging in higher order species (*Richardson et al., 2013*; *Schmidt et al., 2010*; *David-Jürgens and Dinse, 2010*; *Cheng and Lin, 2013*; *Potier et al., 2006*), and also many neurological diseases, such as Alzheimer's disease (*Vico Varela et al., 2019*), migraines (*Vecchia and Pietrobon, 2012*), and epilepsy (*Fritschy, 2008*). Changes in anti-correlativity parallel the increase in neuronal activity in aged worms, measured as a shift toward higher frequency dynamics (*Figure 3a and b*). Interestingly, this shift in frequency power occurs primarily after the loss of anti-correlated signaling, sometime between day 3 and 6 of adulthood. The clinical relevance of such findings is highlighted by a recent study that ties decreased neuron excitability to longevity, both in humans and *C. elegans* (*Zullo et al., 2019*). Our results here indicate that age-associated shifts in the excitatory/inhibitory balance in *C. elegans* are due, not to increased excitatory signaling, but rather a loss of inhibitory signaling.

In *C. elegans*, the presynaptic voltage-gated calcium channel UNC-2/CaV2α has been shown to modulate the excitatory/inhibitory balance of the nervous system. Specifically, *unc-2*(gf) has

a differential effect on excitatory versus inhibitory signaling at the neuromuscular junction, with increased cholinergic signaling and reduced GABAergic signaling (*Huang et al., 2019*). Young *unc-2*(gf) mutants display multiple characteristics similar to those of aged wild-type worms. These mutants initiate spontaneous reversals more frequently than their wild-type counterparts (*Figure 5a*; *Huang et al., 2019*) and also demonstrate an increased sensitivity to aldicarb (a nematode cholinesterase inhibitor), reflecting an increase in neuromuscular acetylcholine signaling (*Huang et al., 2019*). These characteristics are also observed in aged *C. elegans* (*Glenn et al., 2004*; *Podshivalova et al., 2017*; *Mulcahy et al., 2013*). Upon examining young day 1 *unc-2(gf)* worms using our imaging techniques, we observe a breakdown in system organization and shifts in neural dynamics that are strikingly similar to those in wild-type day 9 senescent animals. This is true across all metrics: spectral power (*Figure 5b and c*), neuronal correlativity (*Figure 5d and e*), and system organization as measured by the angular changes in principal component trajectory (*Figure 5f*). Meanwhile, in *unc-2*lf mutants, we observe a partial rescue of such age-associated changes, primarily with respect to spectral power (*Figure 5b and c*).

The UNC-2/CaV2α channel has been directly tied to the removal of GABAergic synapses in dorsal D-type motor neurons during development (*Miller-Fleming et al., 2016*). This process is partially reliant upon calcium-mediated CED-4/Apaf-1 signaling. CED-4/Apaf-1 is a key mediator of the conserved apoptotic caspase pathway which contributes to the removal of presynaptic domains in *C. elegans* (*Meng et al., 2015*; *Miller-Fleming et al., 2016*), synapse elimination in higher order species (*Ertürk et al., 2014*; *Wang et al., 2014*; *Li et al., 2010*), and more substantial neuronal remodeling, such as regeneration following injury (*Pinan-Lucarre et al., 2012*; *Wang et al., 2019*). We find that suppression of the caspase activator CED-4 reduces age-associated decline in neuron dynamics in a manner comparable to that of *unc-2* lf mutation and opposite that of the *unc-2*gfmutation (*Figure 6*). While further investigation will be needed, these results are consistent with a role of UNC-2/CaV2α calcium-mediated caspase activity in the age-associated decline of neuronal signaling.

Huang et al. demonstrated that *unc-2*(gf) leads to a reduction in GABAergic signaling at the neuro-muscular junction. Moreover, they found that these mutant animals were partially resistant to the GABA$_A$ agonist muscimol (*Huang et al., 2019*). Mammalian studies have shown diminished prominence and functional integrity of inhibitory γ-aminobutyric acid (GABA) synapses with age (*McQuail et al., 2015*; *Rozycka and Liguz-Lecznar, 2017*). We find that application of muscimol reshapes the PSD in a manner antithetical to the effects of aging, restoring low-frequency power and reducing high-frequency power in aged day 9 worms. Interestingly, muscimol has a comparable effect on young day 1 animals (*Figure 7a and b*). Muscimol application also partially restores system-wide organization and temporal continuity in day 9 worms (*Figure 7E*). While we do not observe any effects on neuronal correlativity (*Figure 7c and d*), this is unsurprising, as we would expect GABA$_A$ receptors to be agonized across the nervous system irrespective of the underlying neuronal connectivity. In further support of these results we find that truncation of the GABA$_A$ receptor UNC-49 yields mild but contrary effects to that of muscimol. Specifically, in young adults we observe a shift from low- to high-frequency spectral power (*Figure 7a and b*) and a breakdown in system organization and temporal continuity (*Figure 7e*). Overall, these results demonstrate that the changes in neuronal dynamics and activity with age can be effectively reduced by increased GABA signaling, further suggesting that the loss of inhibitory signaling with age contributes to early neuronal decline.

Interestingly, we observed that older animals exhibit increasingly frequent periods of global neuronal quiescence. Behavioral states that meet the criteria of sleep in *C. elegans* have been examined during developmental periods of lethargus (*Nichols et al., 2017*; *Raizen et al., 2008*) and in minimally immobilized young adult animals in which depression of neuronal activity was seen to correlate with sleep-like behavior (*Gonzales et al., 2019*). Here we observed bouts of global neuronal quiescence in older animals similar to those observed during sleep in which only 10–20% of neurons are active. The complete immobilization of the animal in our imaging assays precludes the behavioral measurements necessary to define this state as sleep. Nonetheless, it will be interesting to further investigate how the increased frequencies of sleep-like neuronal states might be linked to the loss of system-wide organization, neuronal hyperactivity, and excitatory/inhibitory imbalance that we observe in older animals.

Finally, we find that several aspects of neuronal dynamics in *C. elegans* are preserved in long-lived *daf-2(e1370)* animals. *daf-2* is a well-studied model for longevity in *C. elegans* that acts through the

silencing of the insulin signaling pathway. Importantly, *daf-2* animals also display extended 'healthspan' with preserved behavioral traits and mobility later in life (*Podshivalova et al., 2017*). Correspondingly, we find that the frequency dynamics of neuronal activity and system-wide dynamics are partially preserved in *daf-2* animals, while neuronal correlativity is not. These findings further emphasize the power of the comprehensive multi-neuron fluorescence imaging we present here to elucidate the breakdown in neuronal signaling and dynamics corresponding to age-associated behavioral decline.

In this study, we have performed the first comprehensive assessment of activity dynamics across a large portion of the aging *C. elegans* nervous system. Our results highlight the conserved similarities with neuronal aging in mammals and further *C. elegans* as a robust model for the study of functional neuronal decline in aging and neurodegenerative diseases. Our unique measurements allow us to link a breakdown in system-wide organization and dynamics to changes in individual neuron activity and signaling. Importantly, we identify a loss of inhibitory signaling as a key element of early aging resulting in disruption of system-wide excitatory/inhibitory neuronal homeostasis. Paralleling mechanisms of synaptic degradation in development, these changes appear to involve UNC-2/CaV2α and are mitigated by application of a GABA$_A$ agonist. With this study, we determine how changes in neuronal signaling and activity drive age-related decline in nervous system dynamics and begin to identify the underlying cellular processes. Our work provides a powerful system for the understanding of the neurological underpinnings of normal aging and neuropathological states.

## Materials and methods

### *C. elegans* strains and maintenance

Hermaphrodite *C. elegans* were maintained at 20°C on nematode growth medium (NGM)-agarose plates coated with food source *Escherichia coli* strain OP50. Experiments were performed on days 1, 3, 6, 9, and 12 of nematode adulthood. Age synchronization was accomplished via timed egg lays, during which gravid adults were allowed to lay eggs on NGM plates for a period of 2 hr, before removal. Adult worms were regularly transferred to fresh plates, as needed, to prevent starvation and to separate the aging adults from their progeny, except when aged on plates containing 50 uM 5-fluoro-2'-deoxyuridine specifically for spontaneous reversal rate behavioral experiments. Confocal microscopy and associated behavioral experiments were performed using the transgenic strain QW1574 *(lite-1[ce314], zfis146[nmr-1::NLSwCherry::SL2::GCaMP6s, lim-4(−3328–2174)::NLSwCherry::SL2::GCaMP6s, lgc-55 (−120–773)::NLSwCherry::SL2::GCaMP6s, npr-9::NLSwCherry::SL2::GCaMP6s] #18.9* [from *zfex696*]), which expresses the nuclear-localized red fluorescent protein NLSwCherry and cytoplasmic calcium-sensitive green fluorescent protein GCaMP6s in eight pairs of premotor interneurons (AVA, AVB, AVD, AVE, AIB, RIM, RIV, and PVC) in an otherwise unmodified wild-type N2 background (*Awal et al., 2018*). When indicated, behavioral and confocal experiments were performed using QW1574 worms that had been crossed with the long-lived strain CB1370 *(daf-2[e1370])*, which harbors a suppressive mutation within the *daf-2* insulin/ insulin-like growth factor 1 receptor ortholog. Light-sheet microscopy and associated behavioral experiments were performed using the transgenic strain QW1217 *(zfis124[Prgef-1::GCaMP6s]; otis355[Prab-3::NLS::tagRFP])*, which pan-neuronally expresses nuclear-localized tagRFP and cytoplasmic GCaMP6s. When indicated, light-sheet microscopy and associated behavioral experiments were performed using QW1217 worms that had been crossed with strain CB1370 *(daf-2[e1370])*, CB407 *(unc-49[e407])*, CB55 *(unc-2[e55])*, QW37 *(unc-2[zf35])*, or MT2547 *(ced-4[n1162])*. CB55 and QW37 harbor lf and gf mutations, respectively, within the CaV2 channel α1 subunit UNC-2. CB407 harbors a premature stop codon within UNC-49, a GABA$_A$ receptor. MT2547 harbors a suppressive mutation within CED-4, a key caspase activator within the conserved apoptotic cell-death pathway and homolog to Apaf-1. To measure tissue autofluorescence among head region neurons with age, we employed the transgenic strain QW1155 *(otis355[Prab-3::NLS::tagRFP])*, which pan-neuronally expresses nuclear-localized tagRFP, but not cytoplasmic GCaMP6s. The following strains and crosses were generated in the laboratory of Dr. Mark Alkema (University of Massachusetts Medical School, Worcester, Massachusetts): QW1574, QW1217, QW1319 *(QW1217; CB55)*, and QW1348 *(QW1217; QW37)*. The following strains were obtained from the Caenorhabditis Genetics Center (University of Minnesota, Minneapolis, Minnesota): CB55 *(unc-2[55])*, MT2547 *(ced-4[n1162])*, and CB1370 *(daf-2[e1370])*.

## Behavior

All *C. elegans* behavior was assessed in a food-free environment.

### Spontaneous reversal rate

Immediately before transfer to food-free NGM-agarose plates, QW1574 or QW1217 strain worms were washed 3× in 1× S-Basal solution (100 mM NaCl, 50 mM KPO$_4$ buffer, 5 µg/ml cholesterol) to remove bacteria. Animals were then left undisturbed for 10 min on clean plates, before assessment. Crawling behavior was video recorded and subsequently analyzed using ImageJ (*Rueden et al., 2017*). The spontaneous reversal rates of individual animals were determined by counting the number of spontaneous reversals initiated during time frames ranging from 2 to 10 min.

### Anterior touch responsiveness

To facilitate the removal of bacteria, QW1574 strain worms were moved to fresh NGM-agarose plates (without bacteria) and allowed to crawl around for 5 min. This was repeated once more, before animals were transferred to a third and final plate, where they were left undisturbed for 10 min prior to behavioral assessments. Individual animals, locomoting in a forward direction, were stroked with an eyelash across the anterior portion of their bodies a total of five times, with a recovery period of at least 5 min between stimuli. Animals were scored as responsive or non-responsive to a given stimulus, if a movement reversal was elicited or not, respectively. Worm populations were blinded to experimenters performing behavioral measurements.

## Confocal microscopy

Confocal fluorescence microscopy was performed as previously described (*Wirak et al., 2020*) to capture spontaneous in vivo activity of the premotor interneuron AVA. In short, worms were briefly paralyzed in 5 mM tetramisole immediately prior to encasement in a transparent and permeable polyethylene hydrogel (*Burnett et al., 2018*). AVA was then imaged using a Zeiss 700 confocal (Carl Zeiss Microscopy, Peabody, MA) with an oil immersion 40× objective. Each trial was 10 min at a capture rate of four images/second and its activity dynamics captured via calcium indicator GCaMP6s fluorescence. Nuclear-localized RFP fluorescence was captured in parallel, to facilitate downstream image registration.

## Light-sheet microscopy

Light-sheet microscopy and imaging preparation were performed as previously described (*Awal et al., 2020*), to capture spontaneous in vivo neural activity within the animal's entire head region at single-cell resolution. Minor changes in protocol include the covalent attachment of the crosslinked polyethylene hydrogel to a silanized glass coverslip (prepared as described; *Burnett et al., 2018*), instead of hydrogel-coverslip attachment via cyanoacrylate ester. Briefly, following hydrogel encapsulation, worms were placed in a Petri dish and immersed in 1× S-Basal solution (100 mM NaCl, 50 mM KPO$_4$ buffer, 5 µg/ml cholesterol) with 5 mM tetramisole. Imaging was performed using a dual-inverted selective plane illumination fluorescence microscope (Applied Scientific Instrumentation, USA) with water immersion 0.8 NA 40× objectives (Nikon USA, Melville, NY). Each animal was imaged for 10 min at a rate of two volumes/second (voxel size 0.1625 × 0.1625 × 1 µm), capturing both nuclear-RFP and cytoplasmic-GCaMP6s fluorescence. Custom Python scripts were employed during postprocessing to track 120 RFP-labeled nuclei per animal in three dimensions and extract GCaMP6s signals from the surrounding somas. These scripts accompany this article as a source code file. While nuclear-RFP fluorescence did decrease with age, the signal was sufficiently greater than background to facilitate equivalent neuron tracking in aged worms (as demonstrated in *Video 3* and *Video 4*). Averaged across worms, the mean fluorescence value of voxels selected as being within nuclei versus that of voxels not selected (i.e. the background) was 155.0 versus 101.5, with a standard error of 5.2 and 0.18, respectively. To measure tissue autofluorescence in the green GCaMP channel among head region neurons with age, we imaged the transgenic strain QW1155 (see *C. elegans* strains and maintenance), which pan-neuronally expresses nuclear-localized tagRFP but not cytoplasmic GCaMP6s, as described above. When indicated, the 1× S-Basal solution in which the animals were immersed was supplemented with 1 mM muscimol.

## Data analysis

All GCaMP6s fluorescence intensity traces were normalized $\Delta F/F_0$ (i.e. the change in fluorescence above baseline, divided by that baseline).

### AVA activity

From the normalized AVA activity traces ($F_0$ defined as the mean value for each trace), calcium transient onsets (i.e. on transitions) and offsets (i.e. off transitions) were manually identified by clear rises and falls in GCaMP6s fluorescence intensity. As previously described (*Wirak et al., 2020*), nonlinear fitting of hyperbolic tangent functions to these on and off transitions facilitated the unbiased analysis of activity state transition properties and eliminated noise. To determine the rise time of a given on transition, its fitted function was used to calculate the time required to rise from 5% of the maximum fluorescence above baseline to 95%. Off transition fall times (i.e. the time required to fall from 95 to 5% of the max fluorescence above baseline) were similarly determined. Average on and off transition plots were generated by aligning individual traces such that the calculated 5% max($\Delta F/F_0$) and 95% max($\Delta F/F_0$) positions, respectively, aligned at the same frame, prior to averaging. To determine the proportion of time that AVA was active versus inactive, its duty ratio was calculated. This was done by summing the calcium transient durations for a given GCaMP6s trace and dividing by the total length of the trace (i.e. 10 min). The duration of a given calcium transient was defined as the time difference between the first frame of its onset and the first frame of its offset.

### Multi-neuron activity

Each imaging condition consists of 10–20 animals, with activity traces measured from 120 neurons per animal via custom image analysis algorithms described in *Awal et al., 2020*. Normalized fluorescence, $\Delta F/F_0$, of all measured neurons was plotted in the activity heatmap for each trial as in *Figure 2* and *Figure 2—figure supplement 2*. For each neuron, $F_o$ was calculated as the mean value of the lowest 1% of measurements made for that neuron. Neuron activity heatmaps were scaled to the dynamic range of values in that trial. Corresponding time correlation heatmaps were derived from the fluorescence measurements in the 120-neuron GCaMP arrays. The relative Euclidean distance in activity space is calculated between every time point. The normalize distance in activity space for two time points is indicated by color: blue represents two time points with very similar neuronal activity patterns while red indicates the largest difference in neuronal activity patterns for that trial. **SNR** measurements were made by manual inspection of activity traces of individual neurons. Following previous studies characterizing GCaMP fluorescence (*Chen et al., 2013*; *Akerboom et al., 2012*), we measured the amplitude of maximal signal (i.e. the largest sustained transient) within the trace and compared it to the mean noise amplitude measured over a significant period (>20 s) of minimal dynamics within the trace to generate the SNR for that neuron (SNR = $20\log_{10}(S/N)$). We examined 12 randomly selected neurons from each animal (i.e. 10% of the 120 neurons imaged in each animal) taken from 10 separate animals at each age (i.e. 120 neurons total per age). Neuron traces were blinded for age and presented at random to the investigator for measurement. The results, displayed in *Figure 2—figure supplement 1*, show no sustained change in fluorescence SNR with age. In an alternative method, we also performed an automated SNR calculation for all neurons (calculated as the maximal signal power measured over a 10 s interval divided by the minimal signal power measured over a 10 s interval for each neuron). Using this method, we again found that mean SNR of day 1 and day 9 animals were not statistically different. **PCA** was performed on the normalized fluorescence traces ($F_0$ defined as the mean of the lowest 1% of values within each trace) and PCA figures generated by plotting the first three principle components on a 3D graph to generate a trajectory over time (PCs were not additionally normalized). Smoothness of these trajectories over time was assessed by calculating the instantaneous change in direction at each time point, that is, the absolute angular difference between the tangential direction over the past 3 s and that of the future 3 s. This calculation amounts to a discrete time derivative of the tangential angle describing the trajectory defined within an intrinsic co-ordinate system and is widely used in analyzing 3D trajectories. Probability histograms for individual animals (trials) or across all trials of a specific condition were generated by pooling the measurements of angular change from all time points within that/those trial(s). Prior to further analyses, time differentials were calculated from normalized activity arrays using total-variation regularization (*Chartrand, 2011*). To calculate mean **PSDs** for each condition (*Figure 3a*), PSDs were generated for each worm's

120-neuron activity array, by Fourier transformation, and then averaged. To account for potential age-dependent changes in GCaMP6s expression and reporter function, these average PSDs were then normalized such that the total power of each is equal to 1. Average PSDs are alternatively displayed as cumulative PSD plots (*Figure 3a*, inset), displaying the proportion of power residing below a given frequency. Relative shifts in PSD power were quantified by calculating for each condition the average 40% spectral edge (i.e. the frequency below which resides 40% of the total spectral power; *Figure 3b*). This was done by averaging each worm's 40% spectral edge. This particular spectral edge was chosen because it demonstrated the most prominent age-dependent shift in wild-type animals. In addition, power spectra were analyzed on an individual neuron basis. The 40% spectral edge was calculated for each individual neuron. All values where pooled across all neurons and all trials for a particular age to generate the spectral edge probability histograms displayed in *Figure 3—figure supplement 2*. **Neuron-neuron correlativity** was calculated as the Pearson correlation between each possible neuron pair among the 40 most dynamically active neurons (i.e. those having the greatest standard deviation in neuronal signal) within a given worm's 120-neuron activity array. This was calculated between the GCaMP6s fluorescence time differentials of the two neurons. The 40 most dynamically active neurons were specifically examined in order to focus on those exhibiting highly dynamic, as opposed to static, activity. A prior study has also shown this fraction to be ~40% among *C. elegans* head region neurons (*Kato et al., 2015*). Neuron pair correlativity probability histograms (*Figure 3c*) were generated for each condition by pooling all neuron-neuron correlativity values across all animals. To determine the proportion of neuron pairs exhibiting strong negative correlativity, the negative correlation proportion (i.e. the proportion of correlativity values less than –0.2; *Figure 3d*) was calculated for each animal and averages generated for each condition. Likewise, to determine the proportion of neuron pairs exhibiting strong positive correlativity, the positive correlation proportion (i.e. the proportion of correlativity values greater than 0.2; *Figure 3e*) was calculated for each animal and averages generated for each condition. Neuron correlativity analysis was repeated with all 120 neurons with similar results (*Figure 3—figure supplement 3*). **Global neural quiescence** was defined in the following way. For each animal/trial a high activity fluorescence value was calculated as the mean value across the 25% of frames with highest fluorescence values. Neurons were considered 'quiescent' if their fluorescence value was below <1/3 of this active fluorescence value. In this way the number of neurons in quiescence could be calculated for each time point and the distribution of neuron quiescence generated for each age (*Figure 4b*). Global quiescence was identified as frames in which >70% of neurons had fluorescence levels below the quiescence cutoff. To demonstrate any effects of intermittent global neuron quiescence on the activity metrics used in this study, we generated two day 9 data subsets: one with significant time spent in global quiescence (17.0% of frames) and one with no time spent in quiescence. The subsets were generated by selecting time segments (consisting of 700 continuous frames each) either with quiescence or without, from the trials taken in old, day 9, animals. These subsets serve to demonstrate the effects of global quiescence on our activity metrics (*Figure 4—figure supplement 1*). Intuitively, increased bouts of global quiescence might increase positive neuron correlativity, as neurons transition into and out of these quiescent states in a concerted manner. While the effect is clear when comparing these day 9 subsets (as shown in *Figure 4—figure supplement 1B*) it is undetectable for the complete day 9 data set (*Figure 3e*) in which global quiescence is less prevalent (~9% time in quiescence for the complete day 9 data set compared to ~17% time in the quiescence data subset). The day 1 data set to which these day 9 subsets are compared has also been truncated to 700 frames for a more consistent comparison (*Figure 4—figure supplement 1*).

## 20 Min trials

To verify that trial length was sufficient to capture the system dynamics and did not alter the measured correlativity, spectral power, or PCA dynamics for old animals that display slowed system-state transitions, we performed 20 min imaging trials with day 9 animals. Results in *Figure 3—figure supplement 4* show minimal difference between 10 and 20 min trials. These longer data sets were further downsampled to ½ and ¼ sampling rates and we verified that this did not alter measured PCA dynamics. **2D-projected videos** of head region neural activity were created for display purposes. For creating each video frame, RFP and GCaMP volumes were flattened to 2D using a maximum intensity projection in the Z-axis (*Wallis and Miller, 1991*). These projections were then combined as the red and green

components of a still RGB image, such that we obtained two frames/second of color imaging. Once all the color still images were generated for the observation period, the image files were streamed into standard video animation software (open-source libx264 codec) to produce the output movie file.

## Statistical methods

One-way ANOVA analyses were initially performed for each metric, across all conditions. Metrics were then compared across all conditions, between figures, with post hoc Sidak multiple comparisons tests. Wild-type data displayed in the figures is the same data set and is displayed versus the relevant data for that figure.

## Acknowledgements

We thank Jennifer Luebke, Monica Driscoll, Harrison Gabel and all members of the Gabel lab for critical discussions related to the paper. We thank Vickery Trinkaus-Randall, Ph.D., director of the Boston University School of Medicine Confocal Microscopy Core, where confocal microscopy experiments were performed (Boston University School of Medicine, Boston, MA). Many strains used in this paper were obtained from the *Caenorhabditis* Genetics Center (University of Minnesota, Minneapolis, Minnesota). Funding was provided by the National Institute for General Medical Sciences (grant R01 GM121457 to C.W.C. and C.V.G. and grant T32 GM008541 to G.S.W.) and the National Institute for Neurological Disorders and Stroke (grant R01 NS107475 to M.J.A.).

## Additional information

### Funding

| Funder | Grant reference number | Author |
| --- | --- | --- |
| National Institutes of Health | NIH R01 GM121457 | Christopher W Connor Christopher V Gabel |
| National Institutes of Health | NIH R01 NS107475 | Jeremy Florman Mark J Alkema |
| National Institutes of Health | NIH T32 GM008541 | Gregory S Wirak |
| Boston University School of Medicine | | Christopher W Connor Christopher V Gabel |

The funders had no role in study design, data collection and interpretation, or the decision to submit the work for publication.

### Author contributions

Gregory S Wirak, Conceptualization, Data curation, Formal analysis, Investigation, Methodology, Writing – original draft, Writing – review and editing, Designed and performed experiments and data analysis. Wrote the manuscript with input from all authors; Jeremy Florman, Resources, generated C. elegans strains and crosses; Mark J Alkema, Resources, generated C. elegans strains and crosses; Christopher W Connor, Formal analysis, Funding acquisition, Software, Writing – review and editing, aided in the analysis of data; Christopher V Gabel, Conceptualization, Formal analysis, Funding acquisition, Investigation, Methodology, Project administration, Supervision, Visualization, Writing – original draft, Writing – review and editing, aided in the analysis of data

### Author ORCIDs

Gregory S Wirak http://orcid.org/0000-0001-9645-1882
Jeremy Florman http://orcid.org/0000-0001-7578-3511
Mark J Alkema http://orcid.org/0000-0002-1311-5179
Christopher V Gabel http://orcid.org/0000-0002-2763-3938

### Decision letter and Author response

Decision letter https://doi.org/10.7554/eLife.72135.sa1

Author response https://doi.org/10.7554/eLife.72135.sa2

## Additional files

### Supplementary files
- Transparent reporting form
- Source code 1. Automated Neuron Tracking Code.

### Data availability
All data generated or analyzed during this study are included in the manuscript and supporting files.

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
