## [Editor Report]

Wirak and colleagues record single cell resolution whole brain dynamics in ageing *C. elegans*. They make the intriguing observations that coordination of brain wide neuronal activity dynamics declines with age, associated with reduced negative correlativity indicating a shift in the excitatory-inhibitory balance across the brain.

---

## [Decision Letter]

**Decision letter after peer review:**

Thank you for submitting your article "Age-associated changes to neuronal dynamics involve a loss of inhibitory signaling *C. elegans* for consideration by *eLife*. Your article has been reviewed by 3 peer reviewers, one of whom is a member of our Board of Reviewing Editors, and the evaluation has been overseen by Piali Sengupta as the Senior Editor. The reviewers have opted to remain anonymous.

The reviewers have discussed their reviews with one another, and the Reviewing Editor has drafted this to help you prepare a revised submission. Reviewers #1 and #2 raise various concerns that your main result could be better explained by any or a combination of following reasons: age associated brain wide quiescence states, a drop in signal-to-noise ratio over age or a consequence of the slowed-down dynamics with increasing age. We believe that addressing these important concerns can mostly be done with additional analyses though new longer recordings might be needed (see reviewer #2). Reviewer #3 is mostly complementary to #1-2 and among others requests the addition of genetic controls. We would be very excited about a revised manuscript in which you can address the following concerns:

Essential revisions:

*Reviewer #1 (Recommendations for the authors):*

(1) The authors report an ageing associated increase in reversal frequency in freely behaving animals, which is not seen in in immobilized animals, however there is an increase in the duration of reversal network states. In lines 6-8, they state that these two findings are congruent, however, I think they should be more cautious with this conclusion. It was shown previously that long reversal command states like shown in Figure 1e or 2a occur only in immobilized animals, while corresponding activity of AVA, AVE, AIB, RIM etc. in freely moving animals appears more spiky and transient (Kato et al., 2015). Reversal frequency and durations in immobilised animals therefore do not necessarily always recapitulate what one can observe in freely behaving animals. This should be more openly discussed in the main text.

(1a) The authors should recapitulate the behavioural experiments in the AVA and whole brain imaging strains. There is a concern that elevated proteostatic stress in transgenic overexpression lines upon ageing could affect neuronal function and behaviour.

(2) The authors should measure how absolute expression levels of GCaMP and the red reference fluorophore change. If GCaMP and/or RFP signals decline over the course of ageing, and/or tissue autofluorescence / background increases, this would cause lower signal noise rations (SNR). The shift in power spectral densities (PSD) as well as in the distribution of angular change in PCA trajectory could be easily explained by a drop in SNR. The authors need to test for these possibilities and provide analyses results showing that there is indeed an increase in activity fluctuations rather than in measurement noise.

(3) The reduction particularly in negative pairwise correlations of regularized time derivatives of the traces is very interesting, but I feel that the interpretation of selective global decline in inhibitory signalling is premature and requires more analyses; particularly since this is one of the major concussions the authors draw from their results.

Typically, in immobilised animals, there are two large clusters of positively correlated neurons, the reversal interneurons + motorneurons and the forward interneurons + motorneurons, plus additional smaller clusters of head motor neurons (Kato et al., 2015). In WT animals, these two major clusters are robustly negatively correlated with each other.

It was shown that immobilization can cause a brain wide quiescence state reminiscent of *C. elegans* lethargus sleep (Gonzales et al., 2019). During sleep, typically all forward neurons become inactive with the exception of RMEs and elevated RIS activity; and animals often revoke from this sleep-like state by initiating reversal states (Nichols et al., 2017). I find it likely that aged worms are more prone to become quiescent in the imaging conditions. If this is the case, the decrease in negative correlations is explained by inactivity of most forward neurons (AVB, RIB, RID, B-MNs within imaging region, etc.) where the animal switches sporadically between a quiescent state and the reverse state. Many of the heatplots in Figure S2 appear to me that this could the case. For their analysis, the authors ranked the activity traces by standard deviation and restricted calculations to the top 40 active neurons. It is therefore likely, that mostly active reversal neurons remained. If this is indeed the case, the major conclusion of the paper falls apart. There would not be a global decline in inhibitory neurotransmission but an increased propensity to enter the quiescent state during forward locomotion, like in (Nichols et al., 2017); the subsequent results from the genetic and pharmacological manipulations are all consistent with this. I, however, think that this would be equally interesting and worth publishing.

If there would be a global decline in inhibitory neurotransmission, as the authors conclude, I would expect that overall activity levels of both forward and backward neurons remain high and that two clusters of positively correlated neuronal ensembles remain. More careful and detailed analysis of activity levels and clustering analyses of the correlation matrices could directly distinguish between the two scenarios. Overall, this is a bit hampered by the lack of cell IDs in the study, acquiring this expertise might be beyond a reasonable time-frame, I think it would tremendously help to ID at least a few neurons like AVA, RIB/AVB, RIS so that forward and backward clusters can be assigned.

(4) The recurrence analyses like in Figure 2b is a nice way to look at these data. It would be fair to credit ref. (Bruno et al., 2017) for introducing this approach to neuronal data analysis. Further, important details are missing in the methods section to properly evaluate this analysis. What is the distance metric used (Euclidean, cosine, correlation…)? Were the PC amplitudes normalized.…? In general, the data analysis section in Methods is very brief and should be expanded.

(5) If I understand correctly, PSDS were calculated across all neurons and then averaged across recordings. This procedure occludes whether PSD distributions reflect diversity of neurons or diversity of signal fluctuations within neuron. This would be interesting to know. More analysis could be done here.

(6) What was F0 for calculating DF/F (mean, min, percdentile?).

(7) In Figure 2a, S2 it looks like traces are min/max normalized; or is this just how heat maps were scaled for display. It should be clearly stated in Figure captions and Methods.

(8) Provide variance explained for PCs and Pareto plots for all PCA analysis.

(9) All data of this study could be shown in supplementary files, like in Figure S2.

(10) It appears that some data are re-utilized across Figures, like WT PSDs of Figure 3a. This is ok but should be clearly started in each Figure panel.

(11) How come that exactly 120 neurons were measured in each recording? Please provide accurate n numbers.

(12) Show SEM or SD for mean traces like PSDs and cumulative power plots.

(13) Line 3-4: I suggest to discuss that aged worm show elevated dwelling like behaviour.

(14) Please use a consistent color & pattern code for genotypes and conditions, see Figure 4 where this gets confused.

Bruno, A.M., Frost, W.N., and Humphries, M.D. (2017). A spiral attractor network drives rhythmic locomotion. *eLife* 6, 471.

Gonzales, D.L., Zhou, J., Fan, B., and Robinson, J.T. (2019). A microfluidic-induced *C. elegans* sleep state. Nature Communications, 1-13.

Kato, S., Kaplan, H.S., Schrödel, T., Skora, S., Lindsay, T.H., Yemini, E., Lockery, S., and Zimmer, M. (2015). Global Brain Dynamics Embed the Motor Command Sequence of *Caenorhabditis elegans*. Cell 163, 1-50.

Nichols, A.L.A., Eichler, T., Latham, R., and Zimmer, M. (2017). A global brain state underlies *C. elegans* sleep behavior. Science (New York, NY) 356, eaam6851.

*Reviewer #2 (Recommendations for the authors):*

The authors nicely demonstrate that neural dynamics slow down as animals age, and that this is also a reflection of known behavioral changes, e.g., enhanced reversals. The changes in the reversal command neurons mimic those behavioral changes, with AVA displaying a larger duty ratio. They also show that neural dynamics are grossly changes in young adult and 9-day old adults. Overall, I believe that this paper should be published in *eLife* if a few major issues are addressed.

(1) To convincingly support their interpretation the authors should add some further sensitivity analyses and control experiments. The key issue for me with the approach taken in this paper is the lack of controls for the changes in the GCaMP signal that can be obtained as a function of age. It is possible that a lot of the effects observed in aging worms is in fact due to changes in protein synthesis and degradation, or changes in the cellular environment altering the kinetics of GCaMP.

(2) – Also related to reviewer #1: But even for clear effects, such as the nicely demonstrated slow-down in neural dynamics the choice of quantification method is not ideal: While the PSD is in principle a good tool for detecting dominant timescales/frequencies in time series, here it might not be the ideal choice: the highly autocorrelated time series (see Figure 2b) only shows very few periods of the lowest frequencies. In this case, the expected output of the PSD is noisy, and dominated by finite sample effects. The presence of 1/f noise is clear from the Figure 3a. To determine how significant the measured PSDs are, error bars would also be very useful.

(3) Similarly, the effects of the smoothness in the PCA trajectories might likely reflect a signal-to-noise ratio issue and a shorter sampling of the neural manifold, rather than a true change in neural activity beyond slowing.

To strengthen these conclusions, the authors could choose one of the following:

– Mimic loss of signal in younger animals e.g. by choosing a less strongly expressed indicator or by bleaching a young adult to a level comparable to day 9 adults.

– Or: determine the expression level of the data.

– _in silico_: test the robustness of the conclusions to noise in the data, as well as shortened sampling. Again, this can be done purely computationally by resampling the day-1 dataset to mimic the slower dynamics of the aged data, and rerun the PSD (to check for finite sampling) and by adding e.g. Gaussian white noise to the dataset (to see the effect of noise of the PCA and correlation metrics).

(4) A key statement made by the authors concerns the loss of correlation between neuron pairs. Due to the rather long autocorrelation within each neuronal trace, the estimation of the time-correlation between neurons is likely biased. This is of concern, since the autocorrelation of neural activity traces between day-1 and day-9 data appear very different, resulting in different effective degrees of freedom. This issue could be addressed by down sampling the day-1 data or -if possible- obtaining a longer day-9 data set and checking if the result still holds.

Note from the consultation session:

I think data of 20 min would definitely help, but one would need to evaluate the actual effective sample number which should be ~Recording length/autocorrelation time and ideally match that between groups.

Other concerns:

– How likely are the neurons that are segmented exactly the same between the day-1 and day-9 worms? Is it possible that due to signal-to-noise different neurons are measured and these have different statistics, affecting the conclusion about the inhibition change?

– Why is df/F0 used when the authors have access to dual-color recordings? It might be advantageous to use the ratiometric signal.

– How is F_0 defined? – see also reviewer #1.

– Figure 4: The data for the positive correlation proportion for ages/genotypes is states as ‘data’ not shown’. It should be displayed somewhere, at least in the supplementary materials.

– The connection between the negative-correlation proportion and excitation-inhibition should be clarified in the Results section and the discussion. It would be especially helpful for the reader if the sentence ‘we would not expect increased inhibitory tone from an exogenous ligand to affect neuronal connectivity’ were unpacked.

*Reviewer #3 (Recommendations for the authors):*

(1) Figure 2: Please add 2D-projected videos of the whole brain activity of day 1 and day 9 worms, for example, as Supplemental Figures.

(2) Figure 2: Please show PC 1-3, like in Figure 1D of Kato et al.’s study (Cell, 2015).

(3) Figure 3: Is it possible to show example(s) of anticorrelated activities by using the heatmap in Figure 2A?

(4) Figure 2: Similarly, is it possible to show example(s) of frequency change in PSD by using the heatmap in Figure 2A? I would like to emphasize this point because, in my understanding, the aged worms exhibited long-term changes in neural activities as seen in Figure 2A and B, which seems inconsistent with the high-frequency shift.

(5) Although the results of genetic and pharmacological analyses are quite interesting, they are unfortunately not convincing. This is because each gene was analyzed using only one material – one mutant allele (unc-2(gf), unc-2(lf), daf-2, ced-4, sel-12) or an agonist (muscimol for GABAA receptor). Therefore, the phenotype could be caused by side mutation(s) or side effects. As a general principle, mutant experiments should be supported by a rescue experiment or by the use of at least two mutant alleles. In the case of unc-2(gf) and unc-2(lf), they seem to exhibit sort of opposite phenotypes, but the results are not strong enough to support the conclusion. In the case of unc-2(gf), N2;Ex[unc-2(gf)] can be used as in the original report (Huang et al., *eLife*).

(6) Furthermore, there is no evidence to support the function of these gene products in the brain, given that relationships between unc-2 and GABA were examined in the motor systems but not in the brain according to two previous studies (Huang et al., *eLife* 2019; Miller-Fleming et al., *eLife* 2016). In my understanding, UNC-2 P/CaV2 is broadly expressed, not only in GABA neurons, therefore it is difficult to see why unc-2(gf) only affects the negatively correlated proportion. Additionally, in which cells/neurons is ced-4 expressed? In ced-4 mutants, the neural circuits could be different because of cell death defects, and to address this issue, cell-type-specific rescue experiment is essential.

[Editors’ note: further revisions were suggested prior to acceptance, as described below.]

Thank you for resubmitting your work entitled “Age-associated changes to neuronal dynamics involve a disruption of excitatory/inhibitory balance in *C. elegans* for further consideration by *eLife*. Your revised article has been evaluated by Piali Sengupta (Senior Editor) and a Reviewing Editor.

The manuscript has been improved but there are some remaining issues that need to be addressed. Please further address the concerns by reviewers #1 and #2 stated below, as well as the minor points by reviewer #3. We may send your second revisions to reviewers #1 and #2 for a final evaluation.

*Reviewer #1 (Recommendations for the authors):*

The authors made major efforts to address my concerns. Still, I have a few remaining issues that I feel are not sufficiently addressed:

(1) Signal-to-noise ratios: I am not convinced that a 12 neuron random sample from each recording is sufficient to assess SNR across all neurons and conditions. I think the authors should devise a systematic SNR analysis across all neurons, which I understand is not trivial but essential.

The authors did not address my concern that expression levels of GcaMP and RFP as well as tissue autofluorescence might change drastically over the course of ageing.

(2) The authors nicely show in the revised manuscript that global quiescence increases with ageing (on its own already an interesting observation) and that this effect does not entirely account for the decrease in anti-correlativity in their wild-type strain. However, this analysis does not exclude that global quiescence does have indeed a contribution to the decrease in anti-correlativity, which could become major in one of their pharmacological and genetic manipulations. The analyses shown in Figure 4b-c and Figure 4 supplement 1 should be performed across all conditions.

*Reviewer #2 (Recommendations for the authors):*

Unfortunately, the authors have not fully responded to my concerns.

My concerns are still regarding the SNR and duration of the measurements. However, I believe these additional analyses could be added in a short timeframe.

1. The definition of SNR used (while previously published), makes the unstated assumption that the noise is independent of the signal amplitude. This is clearly not true here, as can be seen from the raw traces in Figure 4a, and even the single-neuron traces in Figure 1. Sampling the noise of the ‘off’ periods severely underestimates the noise of the measurement and inflates SNR. If the authors could redo the calculations e.g. using fluctuations of a mean-subtracted signal that would be a stronger evidence of no loss of signal as worms age.

2. The 20 minute recordings. While I am happy to hear the authors successfully completed 20 minute recordings, the presented results didn’t cover all my prior questions:

– The PCA plots for these data are not shown, and I would like to see the results of a down-sampling of the 20 min recording to observe the effects of slower neural dynamics on the PCA plot. Similarly, the correlation plot for at least one 20 min recording would be useful to demonstrate the similarity with the 10 min recordings.

– The PSD analysis for the 10 and 20 min data isn’t shown at all.

3. Quiescence analysis

This is a nice addition to the paper, which connects their results to prior studies.

However, I am a bit surprised at how the results are framed: the data in S4.1a (left), and (b) is described as a ‘dramatic reduction’ in negative correlation proportion, whereas the (visually very similar) data in S4.1a (right) is described as not significant. However, the similarly not significant data in S4.1d is described as a trend.

In contrast to the authors, I would interpret these results as at least contributing to their observed effects, given the trends of all measures in Figure S4.1. While a lot of the quiescence effects are not holding up to the small sample size, a n.s. here could be simply due to low sample size. I wonder why the authors did’'t add the 20 minute data here to increase their numbers and possibly get a conclusive answer.

*Reviewer #3 (Recommendations for the authors):*

The authors have adequately addressed most of the previous reviewer comments. I think the manuscript is acceptable for publication once the following minor problems are solved.

1) p. 1, line 10: Unnecessary“"”" in front of“"”".

2) p. 5, line 24: It should be Figure 2-Figure sup. 2, not 1.

3) I am glad to see that unc-49 mutants exhibited the changes opposite to the ones by muscimol. However, day 9 results are not shown in Figure 5-Figure sup. 1, although it was mentioned in the main text (p. 15, line 20-21). Is it a mistake?

---

## [Author Response]

Essential revisions:Reviewer #1 (Recommendations for the authors):(1) The authors report an ageing associated increase in reversal frequency in freely behaving animals, which is not seen in in immobilized animals, however there is an increase in the duration of reversal network states. In lines 6-8, they state that these two findings are congruent, however, I think they should be more cautious with this conclusion. It was shown previously that long reversal command states like shown in Figure 1e or 2a occur only in immobilized animals, while corresponding activity of AVA, AVE, AIB, RIM etc. in freely moving animals appears more spiky and transient (Kato et al., 2015). Reversal frequency and durations in immobilised animals therefore do not necessarily always recapitulate what one can observe in freely behaving animals. This should be more openly discussed in the main text.

We agree that this is an important and relevant distinction to discuss. These prior findings (Kato et al., 2015) are now provided in the Results section.

Related revised manuscript text (and figure):

1. Replaced “…prolonged AVA activation is congruent …” with “…increased AVA activity is consistent…”

2. Added: “However, it should be noted that the neural activity patterns of immobilized *C. elegans* differ from those of freely behaving animals (Kato *et al.*, 2015), making it difficult to draw direct comparisons between the two contexts. Specifically, freely moving animals display short-lived and spikey AVA calcium transients, as opposed to the sustained periods of high activity observed in immobilized animals.”

(1a) The authors should recapitulate the behavioural experiments in the AVA and whole brain imaging strains. There is a concern that elevated proteostatic stress in transgenic overexpression lines upon ageing could affect neuronal function and behaviour.

The behavioral experiments included in this manuscript were exclusively performed using both the AVA imaging strain (Figure 1A-C) and whole brain imaging strain (Figure 5A, Figure 5—figure supplement 2A), as these age-associated changes in behavior have already been well-defined in the literature for wildtype animals. This had been stated in the Materials and methods/*C. elegans* Strains and Maintenance section, however, for added clarity we have now specified the strain used in the Materials and methods/Behavior section and also relevant Figure Legends.

(2) The authors should measure how absolute expression levels of GCaMP and the red reference fluorophore change. If GCaMP and/or RFP signals decline over the course of ageing, and/or tissue autofluorescence / background increases, this would cause lower signal noise rations (SNR). The shift in power spectral densities (PSD) as well as in the distribution of angular change in PCA trajectory could be easily explained by a drop in SNR. The authors need to test for these possibilities and provide analyses results showing that there is indeed an increase in activity fluctuations rather than in measurement noise.

Since submission of our manuscript, we have made a detailed measurement of the signal to noise ratio (SNR) of our GCaMP6s fluorescence measurements at different ages in *C. elegans*. Results, illustrated in what is now Figure 2—figure supplement 1, show no significant decrease in the measured SNR of neuronal fluorescence comparing day 1 to all other ages (day 3, 6, 9, 12 of adulthood). The specific methods for calculating the SNR are now included in the Materials and methods/Data Analysis section.

(3) The reduction particularly in negative pairwise correlations of regularized time derivatives of the traces is very interesting, but I feel that the interpretation of selective global decline in inhibitory signalling is premature and requires more analyses; particularly since this is one of the major concussions the authors draw from their results.Typically, in immobilised animals, there are two large clusters of positively correlated neurons, the reversal interneurons + motorneurons and the forward interneurons + motorneurons, plus additional smaller clusters of head motor neurons (Kato et al., 2015). In WT animals, these two major clusters are robustly negatively correlated with each other.It was shown that immobilization can cause a brain wide quiescence state reminiscent of *C. elegans* lethargus sleep (Gonzales et al., 2019). During sleep, typically all forward neurons become inactive with the exception of RMEs and elevated RIS activity; and animals often revoke from this sleep-like state by initiating reversal states (Nichols et al., 2017). I find it likely that aged worms are more prone to become quiescent in the imaging conditions. If this is the case, the decrease in negative correlations is explained by inactivity of most forward neurons (AVB, RIB, RID, B-MNs within imaging region, etc.) where the animal switches sporadically between a quiescent state and the reverse state. Many of the heatplots in Figure S2 appear to me that this could the case. For their analysis, the authors ranked the activity traces by standard deviation and restricted calculations to the top 40 active neurons. It is therefore likely, that mostly active reversal neurons remained. If this is indeed the case, the major conclusion of the paper falls apart. There would not be a global decline in inhibitory neurotransmission but an increased propensity to enter the quiescent state during forward locomotion, like in (Nichols et al., 2017); the subsequent results from the genetic and pharmacological manipulations are all consistent with this. I, however, think that this would be equally interesting and worth publishing.If there would be a global decline in inhibitory neurotransmission, as the authors conclude, I would expect that overall activity levels of both forward and backward neurons remain high and that two clusters of positively correlated neuronal ensembles remain. More careful and detailed analysis of activity levels and clustering analyses of the correlation matrices could directly distinguish between the two scenarios. Overall, this is a bit hampered by the lack of cell IDs in the study, acquiring this expertise might be beyond a reasonable time-frame, I think it would tremendously help to ID at least a few neurons like AVA, RIB/AVB, RIS so that forward and backward clusters can be assigned.

We thank the reviewer for these astute observations. With additional analysis, we verified that older animals do indeed exhibit periods of sleep-like global neuronal quiescence. The percentage of time spent in a quiescent state (based on a simple metric explained in the methods) increases with age. These results are displayed in Figure 4. These results are highly interesting, and we are grateful to the reviewer for bringing to our attention.

Based on these findings, we further tested how the sleep-like global quiescence effects the metrics of neuronal dynamics we measure across ages. To do this we generated two distinct data sub-sets from the day 9 trials, one that contained significant periods of neuronal quiescence and one with no quiescence. Details are explained in the methods. As both data sub-sets were generated from the same animal trials this allowed for direct comparison of data with and without quiescence. Note the “quiescent” sub-set contained significantly higher percentage of quiescence time points (~17%) compared to that of the complete day 9 data set (~9%). Thus our analysis likely overestimates the effects of quiescence. Nonetheless for the major metrics used in the paper, PCA angular trajectory, PSD spectral edge, Neuron-neuron correlation we found minimal differences between the quiescent and non-quiescent data sub-sets. These results are shown in Figure 4—figure supplement 1. Thus, while the increasing frequency of global quiescence appears to be an important aspect of neuronal aging, it does not explain the large changes in neuronal dynamics that we measure with age.

In addition, we now display neuron correlation data from all 120 neurons measured (rather than the 40 most active neurons), Figure 3—figure supplement 3. As significant drop in anti-correlation with age is also seen in this data set as well.

(4) The recurrence analyses like in Figure 2b is a nice way to look at these data. It would be fair to credit ref. (Bruno et al., 2017) for introducing this approach to neuronal data analysis. Further, important details are missing in the methods section to properly evaluate this analysis. What is the distance metric used (Euclidean, cosine, correlation…)? Were the PC amplitudes normalized.…? In general, the data analysis section in Methods is very brief and should be expanded.

We thank the reviewer for directing us to this reference and have included it in the manuscript. We have added additional detail on how the fluorescence data is displayed in the activity heatmap, time correlation and PCA plots in the methods.

(5) If I understand correctly, PSDS were calculated across all neurons and then averaged across recordings. This procedure occludes whether PSD distributions reflect diversity of neurons or diversity of signal fluctuations within neuron. This would be interesting to know. More analysis could be done here.

We agree that further analysis here would be of interest. Based on this suggestion, we have added additional power spectral analysis based on individual neurons PSD. We calculated the 40% spectral edge for each neuron and then generated a distribution histogram across all neurons measured at a particular age. Results are shown in Figure 3—figure supplement 2 which shows a clear spreading of the distribution toward higher frequencies and corresponding increase in the median 40% spectral edge with age. These results help to confirm the loss of low frequency neuron dynamics which is accompanied by an increase in high frequency dynamics.

(6) What was F0 for calculating DF/F (mean, min, percdentile?).

We have now specified the F0 used for calculating DF/F in the Methods. For each neuron within the multi-neuron arrays, F_o_ was calculated as the mean value of the lowest 1% of measurements made for that neuron in that trial.

(7) In Figure 2a, S2 it looks like traces are min/max normalized; or is this just how heat maps were scaled for display. It should be clearly stated in Figure captions and Methods.

The traces displayed in Figures 2a and what is now Figure 2—figure supplement 2 are ∆F/F0 normalized. Via our response to Reviewer One, Comment #6, F0 is now defined in the Methods. We have also now specified in Figure Legends 2 and Figure 2—figure supplement 2 that these traces are ∆F/F0 normalized.

(8) Provide variance explained for PCs and Pareto plots for all PCA analysis.

We have added a Pareto plot to Figure 2, showing the variance explained by the first five principal components for each age group. We note that directly comparing the system variance explained by each principal component does not reveal clear age-dependent effects.

(9) All data of this study could be shown in supplementary files, like in Figure S2.

To make more data available to the reader, we have added Figure 5—figure supplement 1. For the following conditions, additional data is provided from 3 worms each at days 1 and 9 of adulthood: *unc-2(zf35), unc-2(e55), ced-4(n1162),* muscimol treated, *unc-49(e407), daf-2(e1370).* Here we display for each worm an array of 120-neuron GCaMP fluorescence measurements captured over 10 minutes, alongside the corresponding time correlation heatmaps, the trajectories of the first three principal components, and the probability histograms of the angular directional changes for those principal component trajectories.

(10) It appears that some data are re-utilized across Figures, like WT PSDs of Figure 3a. This is ok but should be clearly started in each Figure panel.

Indeed, we specified in the Methods that: “Wild-type data displayed in the figures is the same data set and is displayed *versus* the relevant data for that figure.” We have now further specified in the relevant figure legends that this is the case.

(11) How come that exactly 120 neurons were measured in each recording? Please provide accurate n numbers.

Our custom Python scripts employed during postprocessing were written to identify and track 120 RFP-labeled nuclei per animal. The method is as described in the section "Statistical Methods for Neuron Tracking and Extraction of Activity" in our earlier publication Awal et al., "Collapse of Global Neuronal States in *Caenorhabditis elegans* under Isoflurane Anesthesia", Anesthesiology 2020 Jul;133(1):133-144. We now directly reference this previous study in the methods. To summarize, we know that there may be individual variation in the expressions of RFP in the neuronal nuclei, and that the total number of observable neurons may vary depending on specimen alignment and the observed region of interest. However, from anatomical considerations and from long experience of making these observations with these specimens, we know that there will never be less than 120 observable nuclei in the recording. The tracking algorithm thus proceeds by first identifying the brightest cluster that is the expected shape and volume of a nucleus. Formally, this is obtained by convolving the three-dimensional dataset with a laplacian-of-gaussian kernel of appropriate construction. The location of the maximum value of this convolution defines the location of the first nucleus of 120 to be chosen. All locations with two nuclear radii of this point are then embargoed for future selection, which prevents selecting any subsequent locations for nuclei that would overlap with those already chosen. The algorithm is then repeated, which thus identifies and selects the second most prominent nuclear expression of RFP as the second nucleus to be chosen. The process is repeated until exactly 120 such nuclear locations are determined. These nuclei are then renumbered in sequential order from nose-to-tail based on their position on the longitudinal axis of the worm.

This approach is more robust than attempting to employ a heuristic to determine the total number of neurons present. If such a counting heuristic were to over-estimate the number of neurons actually present, the detection algorithm would necessarily ultimately exhaust the number of legitimate neurons present and would then begin to attempt to match low-intensity image features that are not neurons. By capping the number of neurons that will be extracted from a particular imaging sequence, this potential source of error is eliminated.

(12) Show SEM or SD for mean traces like PSDs and cumulative power plots.

We agree that it is important to provide the error for power spectral densities and cumulative power plots. Unfortunately, the composite plots with multiple conditions in the main figures become undecipherable with the addition of shaded regions showing measurement error. Thus to maximize visual clarity, we have added a new supplemental figure, Figure 3—figure supplement 1 displaying all relevant pairwise comparisons of PSD plots, both between age and genotype These plots display the standard error of the mean (SEM) and show clearly discernable shifts in the cumulative PSD curves of different ages and experimental conditions.

(13) Line 3-4: I suggest to discuss that aged worm show elevated dwelling like behaviour.

We have now expanded the discussion of age-associated behavioral decline in *C. elegans*, including references to its roaming strategies at the beginning of the Discussion. However, in an effort to keep the focus on the underlying neuronal dynamics we measure here, we do not extend our discussion to include the details of the animal’s complex behaviors.

(14) Please use a consistent color & pattern code for genotypes and conditions, see Figure 4 where this gets confused.Bruno, A.M., Frost, W.N., and Humphries, M.D. (2017). A spiral attractor network drives rhythmic locomotion. eLife 6, 471.Gonzales, D.L., Zhou, J., Fan, B., and Robinson, J.T. (2019). A microfluidic-induced *C. elegans* sleep state. Nature Communications, 1-13.Kato, S., Kaplan, H.S., Schrödel, T., Skora, S., Lindsay, T.H., Yemini, E., Lockery, S., and Zimmer, M. (2015). Global Brain Dynamics Embed the Motor Command Sequence of *Caenorhabditis elegans*. Cell 163, 1-50.Nichols, A.L.A., Eichler, T., Latham, R., and Zimmer, M. (2017). A global brain state underlies *C. elegans* sleep behavior. Science (New York, NY) 356, eaam6851.

For added clarity and to avoid confusion, we have changed the bar chart color corresponding to the unc-2 (gof) genotype from red to yellow.

Reviewer #2 (Recommendations for the authors):The authors nicely demonstrate that neural dynamics slow down as animals age, and that this is also a reflection of known behavioral changes, e.g., enhanced reversals. The changes in the reversal command neurons mimic those behavioral changes, with AVA displaying a larger duty ratio. They also show that neural dynamics are grossly changes in young adult and 9-day old adults. Overall, I believe that this paper should be published in eLife if a few major issues are addressed.(1) To convincingly support their interpretation the authors should add some further sensitivity analyses and control experiments. The key issue for me with the approach taken in this paper is the lack of controls for the changes in the GCaMP signal that can be obtained as a function of age. It is possible that a lot of the effects observed in aging worms is in fact due to changes in protein synthesis and degradation, or changes in the cellular environment altering the kinetics of GCaMP.

As discussed above (response to Reviewer One, Comment #2), we have now performed a rigorous measurement of signal to noise ratio (SNR) of the neuronal fluorescence measurements and find no significant change with age. These results are displayed in Figure 2—figure supplement 1. Details of SNR measurements are described in the methods. The unaltered SNR indicate that the effects we measure with age are indeed the result of altered neuron dynamics rather than due to a breakdown in fluorescence measurement.

(2) – Also related to reviewer #1: But even for clear effects, such as the nicely demonstrated slow-down in neural dynamics the choice of quantification method is not ideal: While the PSD is in principle a good tool for detecting dominant timescales/frequencies in time series, here it might not be the ideal choice: the highly autocorrelated time series (see Figure 2b) only shows very few periods of the lowest frequencies. In this case, the expected output of the PSD is noisy, and dominated by finite sample effects. The presence of 1/f noise is clear from the Figure 3a. To determine how significant the measured PSDs are, error bars would also be very useful.

As discussed in our response to Reviewer 1, Comment #12, we have now included the standard error of the mean for power spectral density (PSD) and cumulative PSD plots. Unfortunately, the composite plots with multiple conditions in the main figures become undecipherable with the addition of shaded regions showing measurement error. Thus, to maximize visual clarity, we have added a new supplemental figure, Figure 3—figure supplement 1 displaying all relevant pairwise comparisons of PSD plots, both between age and genotype. Visual inspection of such plots makes clear the significant separation of conditions we claim to be divergent. Regarding the appropriateness of PSD as a choice of quantification method, we believe that it is of interest to the reader, as it has been employed in other studies to describe neuronal activity (Draguhn and Busáki, 2004) (Busáki and Watson, 2012). Moreover the 1/f noise profile may in fact be an expected property of neuronal networks that feature excitatory/inhibitory responses, as opposed to an unexpected measurement artifact. In a recent paper, this property is demonstrated in a well-studied neural network model (Aguilar-Velázquez and Guzmán-Vargas, 2019).

(3) Similarly, the effects of the smoothness in the PCA trajectories might likely reflect a signal-to-noise ratio issue and a shorter sampling of the neural manifold, rather than a true change in neural activity beyond slowing.To strengthen these conclusions, the authors could choose one of the following:– Mimic loss of signal in younger animals e.g. by choosing a less strongly expressed indicator or by bleaching a young adult to a level comparable to day 9 adults.– Or: determine the expression level of the data.– _in silico_: test the robustness of the conclusions to noise in the data, as well as shortened sampling. Again, this can be done purely computationally by resampling the day-1 dataset to mimic the slower dynamics of the aged data, and rerun the PSD (to check for finite sampling) and by adding e.g. Gaussian white noise to the dataset (to see the effect of noise of the PCA and correlation metrics).

As described in our response to Reviewer One, Comment #2, we have measured the signal to noise ratio of GCaMP6s fluorescence across ages and found no significant decreases. We have further addressed the potential issue raised by shorter sampling of the neural manifold by acquiring longer time-series (doubled from 10 to 20 minutes) with a small sample of worms (n = 5), as described in our response to Reviewer Two, Comment #4. Inspection of these PCA trajectories and their angular change distributions demonstrate that longer datasets do not substantially ameliorate the effects on PCA smoothness observed with age, as would be expected if such effects were purely due to shorter sampling of the neural manifold. Results are displayed in Figure 3—figure supplement 4C.

(4) A key statement made by the authors concerns the loss of correlation between neuron pairs. Due to the rather long autocorrelation within each neuronal trace, the estimation of the time-correlation between neurons is likely biased. This is of concern, since the autocorrelation of neural activity traces between day-1 and day-9 data appear very different, resulting in different effective degrees of freedom. This issue could be addressed by down sampling the day-1 data or -if possible- obtaining a longer day-9 data set and checking if the result still holds.Note from the consultation session:I think data of 20 min would definitely help, but one would need to evaluate the actual effective sample number which should be ~Recording length/autocorrelation time and ideally match that between groups.

The reviewers raise a valid point that the slow dynamics (long duty cycles or substantial autocorrelation) of individual neuron signals could artificially increase the measured correlation between neurons simply by the prolong nature of the neuron states. This effect could then increase with the even slower dynamics in older animals. However, as described briefly in the methods section, our analysis in fact measures the correlations between the *time-derivatives of the fluorescence signals.* We took this approach specifically to mitigate the issues of the highly autocorrelated signals as we are effectively measuring the normalized covariance (*i.e.* the Pearson correlation coefficient) of the change in neuron fluorescence which do not exhibit the prolong stabilized dynamics (autocorrelation) of the fluorescence signals themselves. This analysis follows methods we established in earlier studies (Awal et al: Collapse of global neuronal states in *Caenorhabditis elegans* under isoflurane anesthesia. Anesthesiology 2020; 133(1):133-144) as well as other studies that utilized the time-derivative of the neuron fluorescence as a meaningful metric. In addition, we note that increased autocorrelation in older animals would presumably artificially increase correlation values between neurons and do so equally between correlated and anticorrelated pairs. Instead, we observe stable positive correlation and a loss of anticorrelation with age which is inconsistent with any effects of prolonged signal autocorrelation. Finally, as suggested we have performed longer imaging trials (20 min) in day 9 wild-type animals and find no discernable change in the distribution of correlativity, as compared to the 10 min imaging trials. We have added a new supplementary figure (Figure 3—figure supplement 4), displaying these results. Note: Analyses of system dynamics (PCA) do not employ the time-derivative of the signals as we believe it is conceptually clearer to analyze the fluorescence signals directly where appropriate.

Other concerns:– How likely are the neurons that are segmented exactly the same between the day-1 and day-9 worms? Is it possible that due to signal-to-noise different neurons are measured and these have different statistics, affecting the conclusion about the inhibition change?

It is true that we cannot be certain the neurons segmented for correlation analysis are entirely consistent between the day 1 and 9 datasets. However, as described in our response to Reviewer 1, Comment #3, we demonstrate that the effect on negative correlativity persists with a high degree of significance, even without segmentation, when examining all 120 neurons captured per animal. This is the only metric that involved segmentation of the data. Moreover, as described in our response to Reviewer One, Comment #2, we have measured the signal to noise ratio of GCaMP6s fluorescence across ages and found no significant decreases.

– Why is df/F0 used when the authors have access to dual-color recordings? It might be advantageous to use the ratiometric signal.

We agree that this would be the preferable approach. However, in both imaging strains the GCaMP and RFP employed are expressed as two separate proteins. Therefore, the extent to which the nuclear-localized RFP persists with age may not tightly correlate with the persistence of the cytoplasmic GCaMP. Moreover, in the multi-neuron imaging strain, GCaMP and RFP expressions are driven by different promoters, inserted into separate chromosomes. We therefore cannot rely on ratiometric normalization.

– How is F_0 defined? – see also reviewer #1.

As described above, response to Reviewer One, Comment #6, detail on fluorescence signal analysis has been added to the methods. For each neuron in the multi-neuron arrays, F_o_ was calculated as the mean value of the lowest 1% of measurements made for that neuron in that trial.

– Figure 4: The data for the positive correlation proportion for ages/genotypes is states as 'data' not shown'. It should be displayed somewhere, at least in the supplementary materials.

As requested, we have now added a supplemental figure showing the positive correlation proportion for all ages and genotypes.

– The connection between the negative-correlation proportion and excitation-inhibition should be clarified in the Results section and the discussion. It would be especially helpful for the reader if the sentence 'we would not expect increased inhibitory tone from an exogenous ligand to affect neuronal connectivity' were unpacked.

As requested, we have clarified the connection between negative-correlation proportion and excitation-inhibition, in both the results and Discussion section. We have further unpacked the statement ‘we would not expect increased inhibitory tone from an exogenous ligand to affect neuronal connectivity’ In both results and Discussion section.

Reviewer #3 (Recommendations for the authors):(1) Figure 2: Please add 2D-projected videos of the whole brain activity of day 1 and day 9 worms, for example, as Supplemental Figures.

As requested, we have now provided such videos as supplementary material. A description of how these 2D-projected videos have been created has been added to the Methods.

(2) Figure 2: Please show PC 1-3, like in Figure 1D of Kato et al.'s study (Cell, 2015).

For the examples displayed in Figure 2, we now provide the values of principal components 1-3 over time.

(3) Figure 3: Is it possible to show example(s) of anticorrelated activities by using the heatmap in Figure 2A?

We agree that it would be useful to monitor and display correlativity of particular neuron pairs across ages. However, such analysis would require the identification of particular neurons and neuron pairs across trials and animals, which is beyond our current technical capabilities. Inspection of the 120 neuron activity arrays reveals neuron pairs with varying correlativity and anti-correlativity, which we now emphasize in the text. We have added arrow markers to highlight two such pairs in figure 2a. However, without consistent neuron ID across trials, separately displaying arbitrarily selected pairs to convey relative correlativity at each age does not hold much meaning (and could be somewhat mis-leading). Therefor we rather rely on the ensemble measurements of correlativity for our analysis.

(4) Figure 2: Similarly, is it possible to show example(s) of frequency change in PSD by using the heatmap in Figure 2A? I would like to emphasize this point because, in my understanding, the aged worms exhibited long-term changes in neural activities as seen in Figure 2A and B, which seems inconsistent with the high-frequency shift.

We thank the reviewer for this comment as it is an important point. The PSD measurements show a shift towards higher frequency dynamics with age in Figure 3A,B, yet the system-wide state dynamics actually appear to slow with age in Figure 2A, Figure 2—figure supplement 2. In essence both are true. However, it is the addition of high frequency randomized activity that dominates the dynamics of the individual neurons with age. The system state dynamics, on the other hand, become less prominent and well defined and therefore lose their impact on individual neuron PSD measurements. We have added language in our analysis to emphasize this point. Without the benefit of consistent neuronal identification, we feel in is best to display the ensemble data rather than arbitrarily picking out individual examples.

(5) Although the results of genetic and pharmacological analyses are quite interesting, they are unfortunately not convincing. This is because each gene was analyzed using only one material – one mutant allele (unc-2(gf), unc-2(lf), daf-2, ced-4, sel-12) or an agonist (muscimol for GABAA receptor). Therefore, the phenotype could be caused by side mutation(s) or side effects. As a general principle, mutant experiments should be supported by a rescue experiment or by the use of at least two mutant alleles. In the case of unc-2(gf) and unc-2(lf), they seem to exhibit sort of opposite phenotypes, but the results are not strong enough to support the conclusion. In the case of unc-2(gf), N2;Ex[unc-2(gf)] can be used as in the original report (Huang et al., eLife).

In further support of the observed effects of muscimol being due to GABAA receptor agonization, we now also examine worm strain CB407. In this strain, a premature *unc-49* stop codon yields a truncated UNC-49 GABAA receptor. As expected, the findings in this worm are largely opposite to that of muscimol treatment. That is, in young adults we observe a shift from *low to high* frequency spectral power and a *breakdown* in system organization and temporal continuity. We now have two conditions that support the relevance of GABA signaling (muscimol application and UNC-49 truncation mutation) and two conditions that support UNC-2 channel conductance as being a modulator of the described age-related effects (UNC-2 gain- and loss-of-function mutations). These findings are most central to our study and complement our findings in aged wild-type animals. Additionally, we investigated specific *daf*-2 and *ced*-4 mutants as they are both well-studied and of great interest, due to their strong modulation of pathways relevant to human health. Following findings in the literature (Miller-Fleming et al., 2016) describing a ced-4/unc-2 pathway of synaptic removal, our results with ced-4 serve to support the results with the unc-2 mutations. We therefore believe the involvement of these specific mutants themselves is of sufficient interest but have tempered our interpretation of these results based on the concerns raised by the reviewer. Finally, we have removed the data acquired from *sel-12* mutant worms as these results were peripheral to the central focus of the study on the effects of normal aging.

(6) Furthermore, there is no evidence to support the function of these gene products in the brain, given that relationships between unc-2 and GABA were examined in the motor systems but not in the brain according to two previous studies (Huang et al., eLife 2019; Miller-Fleming et al., eLife 2016). In my understanding, UNC-2 P/CaV2 is broadly expressed, not only in GABA neurons, therefore it is difficult to see why unc-2(gf) only affects the negatively correlated proportion. Additionally, in which cells/neurons is ced-4 expressed? In ced-4 mutants, the neural circuits could be different because of cell death defects, and to address this issue, cell-type-specific rescue experiment is essential.

As the reviewer suggests, UNC-2 appears to be broadly expressed across neurons (McKay et al., 2003) (Hunt-Newbury et al., 2007). While it remains unknown how UNC-2 unilaterally affects negatively correlated neuron pairs, the potential for downstream mechanisms that specifically affect inhibitory signaling exists. This is evidenced by (Huang et al., 2019) who have already shown that the same gain-of-function mutant we employ increases cholinergic neurotransmission among *C. elegans* motor neurons, but decreases GABAergic transmission (accompanied by an increase and decrease in cholinergic and GABAergic synapses, respectively). CED-4 also appears to be broadly expressed across cell types, including the head-region neurons (McKay et al., 2003) (Hunt-Newbury et al., 2007), which makes intuitive sense, as CED-4 is required for programmed cell death initiation (Ellis and Horvitz, 1986). Indeed, as suggested by the reviewer, the *ced-4*(n1162) mutant animal we employ retains additional neurons that had been developmentally fated for cell death (Ellis and Horvitz, 1986). However, we are primarily concerned with the progression of our described effects on neural dynamics *within* a given genotype, with age. Moreover, we directly compare day 1 *ced-4*(n1162) mutants to wild-type animals in Figure 6. There we show minimal effects on the PSD, neuron-pair correlativity distributions, and PCA angular change trajectory distributions, across genotypes.

[Editors' note: further revisions were suggested prior to acceptance, as described below.]

The manuscript has been improved but there are some remaining issues that need to be addressed. Please further address the concerns by reviewers #1 and #2 stated below, as well as the minor points by reviewer #3. We may send your second revisions to reviewers #1 and #2 for a final evaluation.Reviewer #1 (Recommendations for the authors):The authors made major efforts to address my concerns. Still, I have a few remaining issues that I feel are not sufficiently addressed:(1) Signal-to-noise ratios: I am not convinced that a 12 neuron random sample from each recording is sufficient to assess SNR across all neurons and conditions. I think the authors should devise a systematic SNR analysis across all neurons, which I understand is not trivial but essential.

To further support the conclusions of our “manual” SNR estimates, we performed a simple power calculation. In comparing the SNR of day 1 and 9 worms, a random sample of 120 neurons per condition provides sufficient power to conclude that the SNR does not drastically change with age. Given the standard deviations measured in the day 1 and 9 datasets from source data file Figure 2-source data 2, a simple power analysis reveals that changes in mean values of >11% will be detectable between the conditions with 0.95 power or better.

As suggested by the reviewer, we also developed a second automated scheme to measure SNR of all neurons. For each neuron trace, mean subtracted signal power was calculated for all 10 second intervals over the length of the trial. SNR was then estimated for that neuron as the maximal signal power measured (i.e. the largest signal measured) divided by the minimum signal power measured (i.e. the baseline noise with minimal neuronal activity). This effectively automates the process performed manually in the previous SNR analysis. Mean values of SNR (974 and 857, with standard errors of 112 and 59 for days 1 and 9, respectively) are not statistically different (p=0.356), these results are now discussed in the methods. Thus, while GCaMP fluorescence values do decrease with age (see discussion below), we conclude that the SNR of the GCaMP measurements remains robust in aged *C. elegans*.

The authors did not address my concern that expression levels of GCaMP and RFP as well as tissue autofluorescence might change drastically over the course of ageing.

We regret not initially addressing this point directly, as we were focused on validating the overriding SNR concerns. We now discuss the age-related changes in GCaMP and RFP fluorescence levels in the methods. We measure a marked decrease in both the GCaMP and RFP fluorescence with age but in both cases the critical measurements (i.e. GCaMP SNR and cell tracking with RFP) are unaffected. A histogram of the mean GCaMP fluorescence of each neuron across all worms at days 1 and 9 is shown in Figure 2—figure supplement 1C, which has been added to the manuscript. While GCaMP intensities clearly shift toward lower values at day 9 compared to day 1, it is not possible to separate an innate decrease in GCaMP fluorescence (from reduced expression levels for example) from possible changes in neuron activity dynamics. Importantly our analysis above shows that SNR of the GCaMP signal is unaltered with age.

To directly measure the change in green channel tissue autofluorescence with age, we have now imaged worms that express nuclear-RFP but no GCaMP, identified and tracked the neurons, and extracted green autofluorescence from each neuron volume, as described in the manuscript. 10 worms were examined on days 1 and 9 of adulthood. When averaging across worms the mean green autofluorescence extracted from 120 neurons per animal, we find no change between ages. We therefore conclude that there is no significant change in green channel tissue autofluorescence within the head neuron somas. This contrasts with the substantial autofluorescence observed in the gut in older *C. elegans*. Autofluorescence measurements have been added to Figure 2—figure supplement 1.

Fluorescence of nuclear-targeted RFP is employed in the automated image analysis to identify and track individual neurons throughout the trial. While the RFP fluorescence markedly decreases with age, it remains sufficient to facilitate neuron tracking. We calculated for all day 1 and 9 worms the average fluorescence value, across the entire trial, of the voxels that are identified as being within nuclei (i.e. within one nuclear radius of a centerpoint) versus the average value of all other voxels (i.e. the background). These values are plotted in the histograms below. While there is a substantial decrease in nuclear RFP fluorescence intensity with age, the clear distinction between background and RFP labeled-nuclei is maintained. For day 9 animals, the mean fluorescence value of voxels selected as being within nuclei versus that of background voxels was 155.0 versus 101.5, with a standard error of 5.2 and 0.18, respectively. This allows for effective neuron identification and tracking in the aged animals. The mean values of these distributions in the day 9 animals have been added to the Methods. To further demonstrate the reliability of this tracking, we have added Video 3 and Video 4, which show the positions identified as nuclei on top of the original Video 1 and Video 2. These videos show that in both young and aged worms, the positions identified as nuclei closely adhere to the visually obvious nuclear-RFP.

**Author response image 1. sa2fig1:** 

Results

(2) The authors nicely show in the revised manuscript that global quiescence increases with ageing (on its own already an interesting observation) and that this effect does not entirely account for the decrease in anti-correlativity in their wild-type strain. However, this analysis does not exclude that global quiescence does have indeed a contribution to the decrease in anti-correlativity, which could become major in one of their pharmacological and genetic manipulations. The analyses shown in Figure 4b-c and Figure 4 supplement 1 should be performed across all conditions.

As suggested, we have now included in Figure 4—figure supplement 1 the proportion of time head-region neurons exhibit global quiescence, in worms of each genetic and pharmacological condition, at day 1 and 9 of adulthood. These results demonstrate that among these conditions, neural quiescence does not trend with the loss or preservation of neural anti-correlativity.

Reviewer #2 (Recommendations for the authors):Unfortunately, the authors have not fully responded to my concerns.My concerns are still regarding the SNR and duration of the measurements. However, I believe these additional analyses could be added in a short timeframe.1. The definition of SNR used (while previously published), makes the unstated assumption that the noise is independent of the signal amplitude. This is clearly not true here, as can be seen from the raw traces in Figure 4a, and even the single-neuron traces in Figure 1. Sampling the noise of the 'off' periods severely underestimates the noise of the measurement and inflates SNR. If the authors could redo the calculations e.g. using fluctuations of a mean-subtracted signal that would be a stronger evidence of no loss of signal as worms age.

Please refer to the discussion above in the response to Reviewer #1, Comment #1. We now have added further to the SNR analysis of the GCaMP signals with age. As suggested, we have in fact used a mean subtracted calculation of signal power in the SNR analyses. We are confident that our analysis demonstrates robust SNR throughout our experiments.

2. The 20 minute recordings. While I am happy to hear the authors successfully completed 20 minute recordings, the presented results didn't cover all my prior questions:– The PCA plots for these data are not shown, and I would like to see the results of a down-sampling of the 20 min recording to observe the effects of slower neural dynamics on the PCA plot. Similarly, the correlation plot for at least one 20 min recording would be useful to demonstrate the similarity with the 10 min recordings.– The PSD analysis for the 10 and 20 min data isn't shown at all.

Each of these points is now addressed in Figure 3—figure supplement 4. In panel E, we display probability histograms of angular directional changes of PCA trajectory, comparing those of day 1 animals to those of day 9 animals from 10 min and 20 min trials. These distributions are more interpretable than individual PCA plots, as they are representative of worm populations, as opposed to individual animals. We observe minimal effects due to trial length at day 9 compared to the effects measured across ages. In panel A, we display aggregate probability histograms of neuron-pair correlativity, directly comparing the 10 and 20 minute recordings. We find that these distributions tightly overlap, supporting our interpretation that longer recordings do not affect this distribution. We have also now added to the supplementary figure average PSD and cumulative PSD plots, representative of the 20 minute imaging trials, and compare them to those of the 10 minute imaging trials (panel C). We find that these plots are similar and that the 40% spectral edge is not significantly affected by lengthening the datasets (panel D). Finally, we have followed the suggestion of down-sampling the 20 min data sets to ascertain the effects of slower neurodynamics on the PCA plots. Below are the PCA histograms (histogram of angular changes of all PCA trajectories) for the original 20 min data as well at the same data at ½ and ¼ the sampling rates. We ascertain no major shift on the PCA dynamics due to sampling rate. These results are now discussed in the methods.

3. Quiescence analysisThis is a nice addition to the paper, which connects their results to prior studies.However, I am a bit surprised at how the results are framed: the data in S4.1a (left), and (b) is described as a 'dramatic reduction' in negative correlation proportion, whereas the (visually very similar) data in S4.1a (right) is described as not significant. However, the similarly not significant data in S4.1d is described as a trend.In contrast to the authors, I would interpret these results as at least contributing to their observed effects, given the trends of all measures in Figure S4.1. While a lot of the quiescence effects are not holding up to the small sample size, a n.s. here could be simply due to low sample size. i wonder why the authors didn't add the 20 minute data here to increase their numbers and possibly get a conclusive answer.

Our original approach was to compare day 9 GCaMP traces derived from the same individual worms, with and without bouts of global neural quiescence. However, this limited the sample size, as not every worm demonstrated both sufficiently long periods with and without quiescence. We therefore now take the reviewer’s advice and have added additional data to the non-quiescent data set. This consisted of 8 additional GCaMP traces in the day 1 condition, and 6 additional traces in the day 9 (no quiescence) condition. Doing so makes it clear that the overrepresentation of neural quiescence does, as expected, cause an increase in positive neural correlativity (Figure 4—figure supplement 1A right, 1B right). However, this is not observed with age in the absence of global neural quiescence and is not detectable in the complete data sets (Figure 3 C and E) due to the infrequency of quiescence bouts even in older animals. The expanded data sets also provide stronger evidence that the presence of neural quiescence in day 9 animals does not have a substantial effect on spectral power (Figure 4—figure supplement 1C & Figure 4—figure supplement 1D). In addition, the increase in sample size lowered the p-value in comparing the 40% spectral edge of day 1 traces versus day 9 (no quiescence) traces (p = 0.11), providing stronger evidence that aging affects the spectral power distribution in the absence of increased global neural quiescence (Figure 4—figure supplement 1D).

Reviewer #3 (Recommendations for the authors):The authors have adequately addressed most of the previous reviewer comments. I think the manuscript is acceptable for publication once the following minor problems are solved.1) p. 1, line 10: Unnecessary "," in front of ".".

We have deleted the unnecessary comma.

2) p. 5, line 24: It should be Figure 2-Figure sup. 2, not 1.

We now refer to the correct supplemental figure.

3) I am glad to see that unc-49 mutants exhibited the changes opposite to the ones by muscimol. However, day 9 results are not shown in Figure 5-Figure sup. 1, although it was mentioned in the main text (p. 15, line 20-21). Is it a mistake?

We thank the reviewer for catching this inconsistency. This reference to day 9 results was an error, as we only examined *unc-49* animals on day 1 as shown in the figure.